# PAC Reasoning: Controlling the Performance Loss for Efficient Reasoning

## Abstract

Large reasoning models (LRMs) have achieved remarkable progress in complex problem-solving tasks. Despite this success, LRMs typically suffer from high computational costs during deployment, highlighting a need for efficient inference. A popular direction of efficiency improvement is to switch the LRM between thinking and nonthinking modes dynamically. However, such approaches often introduce additional reasoning errors and lack statistical guarantees for the performance loss, which are critical for high-stakes applications. In this work, we propose Probably Approximately Correct (PAC) reasoning that controls the performance loss under the user-specified performance loss tolerance. In particular, we construct an upper confidence bound on the performance loss, formulated as a monotone function of the uncertainty score, and subsequently determine a threshold for switching to the nonthinking model. Theoretically, using the threshold to switch between the thinking and nonthinking modes ensures bounded performance loss in a distribution-free manner. Our comprehensive experiments on reasoning benchmarks show that the proposed method can save computational budgets and control the user-specified performance loss.

## 1 Introduction

Large reasoning models (LRMs) have demonstrated strong performance in tackling complex problem-solving (DeepSeek-AI et al., 2025; Yang et al., 2025a; NVIDIA et al., 2025). However, this strong performance largely depends on long reasoning chains, which substantially increase the computational cost during inference. This phenomenon, often referred to as overthinking (Yue et al., 2025), is evident in mathematical and logic-intensive tasks. And, in applications requiring real-time interaction or large-scale processing, such as text generation (Zhang et al., 2022) and chatbot (Roller et al., 2021), inference efficiency directly determines usability and user experience. Therefore, it is essential to improve the inference efficiency of LRMs.

To address this, existing works proposed to switch the LRM into a nonthinking mode to avoid overthinking (Cheng et al., 2025; Chung et al., 2025; Fang et al., 2025; Li et al., 2025; Liang et al., 2025; Ma et al., 2025; Paliotta et al., 2025; Pan et al., 2025; Xiao & Gan, 2025; Yong et al., 2025; Yue et al., 2025). While effective in reducing computational demands, using a nonthinking model often degrades solution quality or introduces additional errors. For instance, in theorem-proving tasks, switching techniques may lead to invalid logical steps, and in mathematical reasoning, it can result in calculation mistakes or overlooked solution paths. Besides, such methods lack a rigorous theoretical guarantee for performance loss. This limitation raises a fundamental issue:

*How to improve the efficiency of LRMs, guaranteeing the performance loss?*

In this work, we formalize this challenge by introducing the concept of a PAC efficient model, a notion where an LRM provides statistical guarantees that its performance loss stays within a user-specified tolerance, as defined in Definition 1. To meet this requirement, we propose **PAC reasoning**, which constructs a composite model $\hat{f}$ that selectively switches between the thinking model $f$ and its cheaper non-thinking counterpart $\tilde{f}$. Concretely, PAC reasoning determines a switching threshold on a calibration dataset via a PAC calibration procedure (Algorithm 1), and during testing (Algorithm 2), it accepts the output of $\tilde{f}$ if its uncertainty score is below the threshold; otherwise, it

resorts to $f$ for reliability. In this way, PAC reasoning complements existing switching approaches by improving efficiency while offering statistical guarantees on performance loss.

Theoretically, we show that PAC reasoning achieves distribution-free control of performance loss with probabilistic guarantees. We formalize this through the composite model $\hat{f}$, whose key property is that the loss function is monotonic in the uncertainty score. This monotonicity enables the construction of an upper confidence bound on performance loss and the derivation of a valid threshold. Under mild regularity conditions, we prove that PAC reasoning keeps the loss below the user-specified tolerance, thereby satisfying PAC efficiency.

We then present comprehensive experimental results [1] in Section 4 that rigorously evaluate the PAC reasoning across diverse reasoning benchmarks, including MATH-500 (Lightman et al., 2023), ZebraLogic(Lin et al., 2025), and Arena-Hard (Li et al., 2024). The results demonstrate that our approach effectively controls the performance loss and significantly reduces inference cost. For example, on Arena-Hard with performance tolerance $\epsilon = 0.08$ for the logits uncertainty score, our method controls the average empirical performance loss at $0.06$ (below the tolerance), and achieves token savings exceeding $40\%$. We also find that the logits-based uncertainty score provides more stable performance loss control compared to the verbalized-based score.

Our contributions are as follows:

- We introduce the concept of $(\varepsilon, \alpha)$-**PAC efficient**, a novel notion for quantifying performance loss of efficiency improvement under PAC-style guarantees in LRMs, and to the best of our knowledge, the first formal such guarantee in this setting.

- We propose **PAC reasoning**, a method that combines a thinking-mode model with its nonthinking counterpart via an uncertainty-based mechanism to improve efficiency. The method is model-agnostic and provides *distribution-free* performance guarantees.

- We provide comprehensive experiments on mathematical reasoning, logical deduction, and text generation, demonstrating that PAC reasoning achieves efficiency gains while satisfying the statistical validity of the PAC efficient guarantee.

**Notations** We begin by introducing key notations. The first is the LRM with thinking-mode $f$, which is computationally expensive but delivers high performance on its answers. Given an input prompt $x$, $f$ produces an output $y = f(x)$, which we regard as the "expert answer". The second is the nonthinking LRM $\tilde{f}$, which is computationally cheaper but potentially less accurate, and $\tilde{y} = \tilde{f}(x)$. And for any input $x$, we use $y^{gold}$ as its "gold reference". Let $\mathcal{I}_{cal} = \{1, \ldots, n\}$ and $\mathcal{I}_{test} = \{n+1, \ldots, n+N\}$ denote the indices of the calibration and test sets, respectively. Accordingly, we define the calibration dataset and the test dataset as:

$$\mathcal{D}_{cal} = \{(x_i, y_i)\}_{i \in \mathcal{I}_{cal}}, \quad \mathcal{D}_{test} = \{(x_i, y_i)\}_{i \in \mathcal{I}_{test}},$$

where each $x_i$ is an input prompt. It is worth noting that the $y_i$ is not a ground-truth label, but the "expert answer" provided by the LRM $f$. Finally, let $y_i = (y_{i,1}, \ldots, y_{i,l_{y_i}})$ denote an answer consisting of $l_{y_i}$ tokens, with $y_{i,j}$ representing the $j$-th token of $y_i$.

## 2 PROBABLY APPROXIMATELY CORRECT REASONING

### 2.1 PAC EFFICIENT MODEL

We aim to build a more efficient LRM, denoted by $\hat{f}$, that provides probably approximately correct guarantees for its performance loss while improving efficiency. Specifically, given an error tolerance $\epsilon$ and a confidence level $1 - \alpha$, $\hat{f}$ ensures its performance loss relative to the thinking-mode LRM $f$ does not exceed $\epsilon$ with probability at least $1 - \alpha$. We formulate the PAC guaranteed $\hat{f}$ as follows:

---

[1] The reproducibility code is placed at an anonymous link.

**Definition 1** $((\epsilon, \alpha)$-PAC efficient). An LRM $\hat{f}$ is called an $(\epsilon, \alpha)$-probably approximately correct (PAC) efficient model (with respect to loss $\ell$) [2], if for given $\epsilon > 0, \alpha \in (0, 1)$ it satisfies

$$\mathbb{P}\left(R(\hat{f}) \leq \epsilon\right) \geq 1 - \alpha,$$

where $R(\hat{f}) = \mathbb{E}_{x \sim P}[\ell(\hat{f}(x), f(x))]$ is the risk function, $\ell(\cdot, \cdot)$ is a loss function, $x$ denotes an input prompt drawn from the underlying task distribution $P$.

*Remark* 2. Here, the loss function can be a 0-1 loss for verifiable tasks or a semantic loss for generative tasks. The positive value $\epsilon > 0$ is called the error tolerance, and $1 - \alpha$ is termed the confidence level. We sometimes term $(\epsilon, \alpha)$-PAC efficient model simply as PAC efficient model.

## 2.2 PAC REASONING

Constructing such a controllable LRM $\hat{f}$ is straightforward intuitively. Given an LRM with thinking mode $f$ and a fast LRM without thinking $\tilde{f}$, we create an intermediate model that selectively uses either the LRM with thinking or not based on certain conditions. This condition acts like a "sliding rheostat" that allows us to tune the performance trade-off by adjusting the "position" of the intermediate. We can obtain a model $\hat{f}$ that achieves the desired error tolerance heuristically. However, this approach lacks statistical guarantees on the underlying distribution of performance loss. To build a model with statistical guarantees, a hypothesis test will be used to determine an optimal threshold that balances computational efficiency with output quality while maintaining statistical confidence.

Motivated by this, we present the **PAC reasoning**, which constructs a composite LRM $\hat{f}$ that improves the efficiency of an LRM with thinking $f$. The composite LRM $\hat{f}$ provides PAC guarantees for the efficiency improvement. The core idea is to use **uncertainty scores** to build an **upper confidence bound** for the performance loss. We could use the upper confidence bound to measure the uncertainty of performance loss for each value of the uncertainty score. Then we **calibrate** an uncertainty threshold to switch between the thinking and nonthinking models. We use the nonthinking LRM $\tilde{f}$ on most inputs and strategically invoke the expensive LRM with thinking $f$ only for inputs whose generation by $\tilde{f}$ has high uncertainty. Next, we provide the details of the PAC reasoning.

### 2.2.1 UNCERTAINTY SCORES AND CUMULATIVE ERROR

We assume that for each input prompt $x_i$, the nonthinking LRM $\tilde{f}$ produces an output $\tilde{y}_i$, and that there exists a score $U_i \in [0, 1]$ to quantify its uncertainty. This score should ideally correlate with the likelihood of disagreement with the reference model $f$. The core idea is to use these uncertainty scores to selectively use the expensive model with thinking $f$. We aim to find a threshold, $\hat{u}$, and accept the nonthinking LRM's output $\tilde{y}_i$ for the instances such that $U_i < \hat{u}$, while querying the model with thinking $f$ for the cases where $U_i \geq \hat{u}$. To formalize, we define the cumulative error function conditioned on the uncertainty threshold $u$:

$$L(u) = \frac{1}{N} \sum_{i=n+1}^{n+N} \ell(y_i, \tilde{y}_i) \mathbf{1}\{U_i \leq u\}. \tag{1}$$

This function measures the average error for test data points with uncertainty scores no greater than $u$. If we could compute $L(u)$ for all $u$, we would choose the largest threshold $u^*$ such that $L(u^*) \leq \epsilon$. However, computing $L(u)$ requires access to all expert answers $y_i = f(x_i)$ in the test set, which is computationally expensive. We try an alternative way to build a bound $\hat{L}_u(\alpha)$ for $\mathbb{E}L(u)$ satisfying the following inequality pointwise w.r.t. $\alpha$:

$$\mathbb{P}(\hat{L}_u(\alpha) \geq \mathbb{E}L(u)) \geq 1 - \alpha. \tag{2}$$

Because the cumulative error function is monotone, we can easily obtain the PAC guarantee as in Definition 1. The monotonicity is a key property in our method, and it allows us to test fixed-sequences single-start without additional corrections (Angelopoulos et al., 2025b); we discuss it in Appendix C, and summarize it in Assumption 3.1.

---

[2] For simplicity, we often omit mentioning "with respect to $\ell$" since most tasks have their conventional loss functions

---

**Algorithm 1** Compute Confidence Bound $\hat{L}_u(\alpha)$ Based on Central Limit Theorem

---

**Input:** Calibration set $\{(x_i, y_i)\}_{i=1}^n$, model with thinking $f$, model without thinking $\tilde{f}$, uncertainty scores $\{U_i\}_{i=1}^n$, sampling weights $\{\pi_i\}_{i=1}^n$, sampling size $m$, and confidence level $\alpha$.

**Output:** The confidence upper bound $\hat{L}_u(\alpha)$.

1: Initialize an empty list of samples $\mathcal{Z} = []$.
2: Let $\tilde{y}_i = \tilde{f}(x_i)$ for all $i = 1, \dots, n$.
3: **for** $j = 1, \dots, m$ **do**
4:     Sample an index $i_j \sim \mathrm{Unif}(\{1, \dots, n\})$.
5:     Sample a Bernoulli random variable $\xi_{i_j} \sim \mathrm{Bern}(\pi_{i_j})$.
6:     **if** $\xi_{i_j} = 1$ **then**
7:         Query the true label $y_{i_j}$ and compute the importance-weighted loss $Z_j = \ell(y_{i_j}, \tilde{y}_{i_j})/\pi_{i_j}$.
8:     **else**
9:         $Z_j = 0$.
10:    **end if**
11:    Append $Z_j$ to $\mathcal{Z}$.
12: **end for**
13: For a threshold $u$, define the variables $Z_j(u) = Z_j \cdot \mathbf{1}\{U_{i_j} \leq u\}$ for $j = 1, \dots, m$.
14: $\hat{\mu}_Z(u) \leftarrow \frac{1}{m} \sum_{j=1}^m Z_j(u)$
15: $\hat{\sigma}_Z(u) \leftarrow \sqrt{\frac{1}{m-1} \sum_{j=1}^m (Z_j(u) - \hat{\mu}_Z(u))^2}$
16: $z_{1-\alpha} \leftarrow (1-\alpha)$-quantile of the standard normal distribution.
17: **Return** $\hat{L}_u(\alpha) \leftarrow \hat{\mu}_Z(u) + z_{1-\alpha} \frac{\hat{\sigma}_Z(u)}{\sqrt{m}}$.

---

### 2.2.2 CONSTRUCTING THE UPPER CONFIDENCE BOUND (UCB)

To compute a feasible average error approximation $\hat{L}_u(\alpha)$, we propose a procedure inspired by probably approximately correct labeling (Candès et al., 2025). Given a sampling size $m$, we first collect $m$ indices $\{i_1, \dots, i_m\}$ by sampling uniformly with replacement from $\{1, \dots, n\}$. Then, for each selected index $i_j$, we decide whether to query its expert answer $y_{i_j}$ by performing a Bernoulli trial $\xi_{i_j} \sim \mathrm{Bern}(\pi_{i_j})$, where $\{\pi_1, \dots, \pi_n\}$ are sampling weights. This procedure yields a dataset of $m$ i.i.d. random variables:

$$Z_j(u) = \ell(y_{i_j}, \tilde{y}_{i_j}) \frac{\xi_{i_j}}{\pi_{i_j}} \mathbf{1}\{U_{i_j} \leq u\}. \tag{3}$$

The expectation of $Z_j(u)$ is equals the target quantity $L(u)$, since $\mathbb{E}[\xi_{i_j}/\pi_{i_j}|i_j] = 1$. We can therefore estimate an upper bound for $L(u)$ by computing a confidence interval for the mean of $\{Z_j(u)\}_{j=1}^m$. We formally described it in the central limit theorem (CLT) based Algorithm 1.

*Remark* 3. The procedure in Algorithm 1 uses importance sampling to construct an unbiased estimator for the true error $L(u)$. For any fixed threshold $u$, the random variables $Z_j(u)$ are i.i.d. with expectation $\mathbb{E}[Z_j(u)] = L(u)$. This holds because the sampling process decouples the choice of index $i_j$ from the decision to query the label $y_{i_j}$. Given the samples $\{Z_j(u)\}_{j=1}^m$, we can form an upper bound for $L(u)$. Algorithm 1 illustrates this using a CLT-based approach, valid for large $m$. See Appendix D for discussion about its UCB validation as Assumption 3.1. Alternatively, if the importance-weighted losses are bounded, one could use concentration inequalities like Hoeffding's or Bernstein's inequality to construct a valid confidence bound (Bentkus, 2004; Hao et al., 2019; Hoeffding, 1994; Learned-Miller & Thomas, 2020; Ramdas et al., 2022; Waudby-Smith & Ramdas, 2021; 2024), which may provide better guarantees for smaller sample sizes. We give an example in Algorithm 3 in Appendix F based on Hoeffding's inequality (Hoeffding, 1994).

### 2.2.3 CALIBRATION

Once the UCB $\hat{L}_u(\alpha)$ is constructed, the threshold $\hat{u}$ is the highest uncertainty level for which this estimated error bound remains below the tolerance $\epsilon$:

$$\hat{u} = \max\{u \in [0, 1] : \hat{L}_u(\alpha) \leq \epsilon\}. \tag{4}$$

We calibrate the $\hat{u}$ on the calibration set $\mathcal{D}_{cal}$, and apply it to the test sample. If the uncertainty score is larger than $\hat{u}$, we use the thinking model to answer; otherwise, we use the nonthinking mode. This

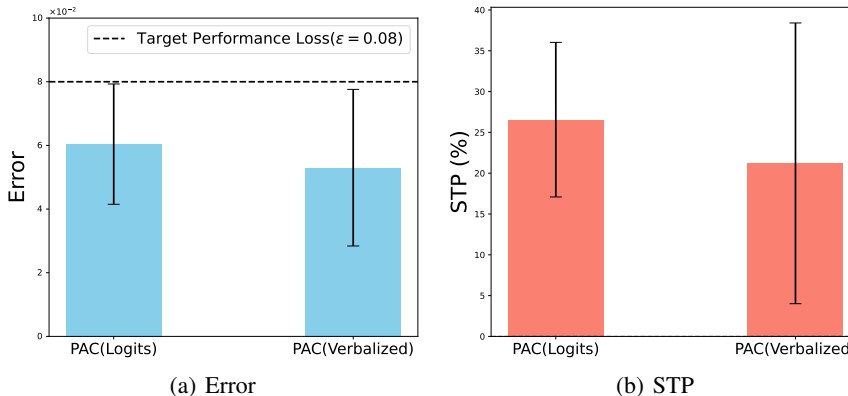

(a) Error                                    (b) STP

Figure 1: **Error control and saved token percentage (STP) of PAC reasoning,** with binary loss on ZebraLogic at a confidence level $95\%$. "PAC(Logits)" and "PAC(Verbalized)" present PAC reasoning using the logits-based score and the verbalized score. Both control the performance loss under the target $0.08$ and save at least $20\%$. All experiments are repeated 100 times, and other details follow as the main experiment in Section 4.1 and Appendix K.

procedure ensures we can accept as many outputs from the nonthinking model $\tilde{f}$ while controlling the overall performance loss with high probability. We summarize the PAC reasoning algorithm in Algorithm 2. And we show the error and saved token percentage on the dataset "ZebraLogic" using PAC reasoning with logits-based score and verbalized score in Figure 1. Our method controls the performance loss below the target loss $0.08$ and saves token cost at least $20\%$ with confidence $95\%$. As for the naive method discussed in Appendix M, switching model using a fixed threshold, it could not control the performance, and may incur more token usage, see details in Table 3.

---

**Algorithm 2** PAC Reasoning

---

**Input:** Calibration set $\{(x_i, y_i)\}_{i=1}^n$, test prompts $x_i, i \in \mathcal{I}_{test}$, model without thinking $\tilde{f}$, model with thinking $f$, loss function $\ell$, error tolerance $\epsilon$, confidence level $\alpha$

**Output:** PAC reasoning model $\hat{f}$ on test set $\mathcal{I}_{test}$

1: $\forall i \in \mathcal{I}_{cal}$, compute outputs without thinking $\tilde{y}_i = \tilde{f}(x_i)$ and uncertainty scores $U_i$.
2: Compute confidence bound function $\hat{L}_u(\alpha)$ via Algorithm 1 on calibration data $\{(x_i, y_i)\}_{i=1}^n$.
3: Determine threshold $\hat{u} = \max\{u \in \{U_i\}_{i \in \mathcal{I}_{cal}} : \hat{L}_u(\alpha) \leq \epsilon\}$.
4: $\forall i \in \mathcal{I}_{test}$, compute outputs without thinking $\tilde{y}_i = \tilde{f}(x_i)$ and uncertainty scores $U_i$ independently.
5: $\hat{y}_i \leftarrow f(x_i)\mathbb{1}\{U_i \geq \hat{u}\} + \tilde{y}_i\mathbb{1}\{U_i < \hat{u}\}$ for all $i \in \mathcal{I}_{test}$.
6: Define the composite model $\hat{f}$ by its outputs: $\hat{f}(x_i) = \hat{y}_i$ for all $i \in \mathcal{I}_{test}$.
7: **Return** $\hat{f}$.

---

## 3 THEORETICAL ANALYSIS

In this section, we aim to introduce the PAC guarantee. Our PAC reasoning builds upon the theoretical foundation established by the distribution-free risk control framework (Angelopoulos et al., 2025b). As discussed in Appendix B, our PAC reasoning problem can be viewed as a generalization of the distribution-free risk control method. It provides the mathematical foundation for our risk control type method on the characteristics of PAC reasoning. In detail, if the performance loss $L(u)$ is bounded by a UCB, and the UCB based on CLT or concentration inequality is valid as Assumption 3.1, we can prove the PAC guarantee as follows.

## 3.1 PAC GUARANTEE

First, noting that while the confidence bound $\hat{L}_u(\alpha)$ is constructed to hold for a single, pre-specified threshold $u$, our algorithm selects the threshold $\hat{u}$ based on the calibration data. Let $\mathcal{D}_{cal}$ be a calibration set, used to construct a threshold $\hat{u}$, and let $\mathcal{D}_{test}$ be an independent test set with i.i.d. samples as $\mathcal{D}_{cal}$. For any threshold $u$, recalling the deployment strategy $T_u(x)$ as Eq. (6), we could see that $\hat{f} = T_{\hat{u}}$ is the composite model. We re-parameterize $\hat{f}$ and its risk with respect to $u$, with a slight abuse of notation. Its population risk $R(\hat{f})$ is re-parameterized as:

$$R(u) = \mathbb{E}[\ell(y, T_u(x))],$$

and the empirical risk is re-parameterized as:

$$\widehat{R}(u) = \frac{1}{N} \sum_{i \in \mathcal{I}_{test}} \ell(y_i, T_u(x_i)).$$

Then we list the assumptions of the PAC reasoning.

**Assumption 3.1** (UCB validity). *For each threshold $u$ and any $\alpha \in (0, 1)$, there exists a UCB $\widehat{L}_u(\alpha)$, computed on $\mathcal{D}_{cal}$, such that*

$$\mathbb{P}\big(R(u) \leq \widehat{L}_u(\alpha)\big) \geq 1 - \alpha.$$

As discussed in Remark 3, we can build the UCB $\widehat{L}_u(\alpha)$ for $\mathbb{E}L(u)$ in two main ways: using the central limit theorem, or using bounds like Hoeffding's or Bernstein's inequality (Bentkus, 2004; Hoeffding, 1994; Waudby-Smith & Ramdas, 2021; 2024), which may provide better guarantees for smaller sample sizes. We prove the validity of the CLT-based method in Appendix D. The risk function $R(u)$ is naturally non-decreasing with $u$. As $u$ increases, the condition $U(x) \geq u$ becomes harder to satisfy, so we defer to the expert less often. Then, reducing deference to the expert can only increase the total risk under Assumption 3.1. We gather the above assumptions and the monotonicity, and provide the PAC guarantee of our proposed method:

**Theorem 4** (PAC guarantee). *Let $\hat{u}$ be the threshold selected by the PAC reasoning algorithm (Algorithm 2). If calibration set and test set are i.i.d. and Assumption 3.1 holds, then the composite model $\hat{f}$ constructed by Algorithm 2 satisfies the $(\epsilon, \alpha)$-PAC guarantee, i.e.,*

$$\mathbb{P}(R(\hat{f}) \leq \epsilon) \geq 1 - \alpha.$$

We prove it in Appendix C. If the loss is bound in $[a, b]$, we provide an empirical version:

**Theorem 5** (Empirical risk PAC guarantee). *Assume Assumption 3.1 holds, and the test batch $\mathcal{D}_{test}$ is independent of the calibration data $\mathcal{D}_{cal}$. Given $\epsilon, \alpha \in (0, 1)$, $\ell \in [a, b]$, and $\hat{u}$ defined as in Theorem 4, then any $t > 0$,*

$$\mathbb{P}\big(\widehat{R}(\hat{u}) \leq \epsilon + t\big) \geq 1 - \alpha - \exp\Big(-\frac{2Nt^2}{(b-a)^2}\Big).$$

*Remark* 6. A common special case is a bounded loss $\ell \in [0, 1]$, e.g., 0-1 loss for binary verifiable answers. Then $b - a = 1$ and the bound simplifies to $\mathbb{P}\big(\widehat{R}(\hat{u}) \leq \epsilon + t\big) \geq 1 - \alpha - e^{-2Nt^2}$. It provides exact risk control for $\hat{f}$ with probability at least $1 - \alpha$ by some slacks $t$.

We prove it in Appendix E. If the i.i.d. assumption does not hold, PAC reasoning can be extended to a transductive setting. This extension is discussed in Appendix G.

## 4 EXPERIMENTS

In this section, we present the experimental results that evaluate the performance loss and computational savings of the proposed PAC reasoning across diverse benchmarks, including mathematical reasoning (Lightman et al., 2023), logical deduction (Lin et al., 2025), and text generation tasks (Li et al., 2024). We aim to verify whether the proposed method could control the performance loss with a confidence level while saving computational resources. We evaluate PAC reasoning under different uncertainty estimators (logits-based and verbalized uncertainty score), as well as efficiency

metrics (expert calling percentage and saved token percentage). Experimental setup details are provided in Section 4.1. Across diverse benchmarks, PAC reasoning consistently controls performance loss under the desired tolerance while saving computation, regardless of the uncertainty estimator or loss function, as shown in Figure 2 and Table 3. We also discuss the behaviors of PAC reasoning and a naive method among different loss functions, uncertainty scores in Section 4.2. We conduct additional experiments on more datasets, GPQA (Rein et al., 2024) and HumanEval (Chen et al., 2021) in Appendix O.4 and comparision with three other method, "naive contorl", Chain of Draft (CoD) (Xu et al., 2025) and NoThinking (Ma et al., 2025), in Appendix O.5.

## 4.1 SETUP

**Large language models**   In this study, we evaluate the PAC reasoning based on the Qwen3 series models (Yang et al., 2025a). Specifically, we employ the "Qwen3-4B-Thinking-2507" as the thinking model and "Qwen3-4B-Instruct-2507" as the lower-performance nonthinking model. The sampling temperature and other hyperparameters for both LLMs are configured following the settings in the original paper. Details can be found in Appendix K. We also conduct a complementary experiment on Llama-3.1-8B–based models, the "DeepSeek-R1-Distill-Llama-8B" as the thinking model and "Llama-3.1-8B-Instruct" as the lower-performance nonthinking model, with results shown in Appendix N.

**Uncertainty score**   We adopt two complementary perspectives to quantify the uncertainty of $\tilde{y}$: a white-box score derived from model logits and a black-box score obtained from verbalized self-reports. We leverage token-level probabilities computed from the prediction logits (Kwon et al., 2023; Zheng et al., 2024; Zhou et al., 2025) for the white-box score. Furthermore, we define the uncertainty score of $y_i$ as its average token probability (Hao et al., 2023; Huang et al., 2025):

$$U_{logits}(y_i) = 1 - \frac{1}{l_{y_i}} \sum_{j=1}^{l_{y_i}} \mathbb{P}(y_{i,j}|y_{i,1}, \ldots, y_{i,j-1}, x_i),$$

where $\mathbb{P}(y_{i,j}|y_{i,1}, \ldots, y_{i,j-1}, x_i)$ is the conditional probability of token $y_{i,j}$. Moreover, we also consider verbalized uncertainty scores from nonthinking models (Xiong et al., 2023; Tian et al., 2023; Yang et al., 2024; Zhou et al., 2025), where the model explicitly states its self-reported confidence. The verbalized uncertainty score is mainly applicable in black-box scenarios, where access to generation logits is restricted, especially in the case of closed-source LLMs. In this study, we report the average confidence over 10 trials, and the corresponding prompts are listed in Appendix K.

**Datasets**   We evaluate PAC reasoning on a series of real datasets spanning reasoning and open-ended generation tasks. Specifically, our evaluation covers a high-level mathematics benchmark, MATH-500 (Lightman et al., 2023), a text-based logical reasoning task, ZebraLogic (Lin et al., 2025), and an alignment-focused open-ended writing benchmark, Arena-Hard (Li et al., 2024). For each dataset, we partition the original test set into a calibration subset and a held-out test subset randomly. Table 2 in Appendix K provides details on the specific splitting strategies.

**Loss functions**   We consider two types of loss functions for evaluating the PAC guarantee of our method: the semantic cosine distance and binary 0-1 loss. Details are placed in Appendix K.

**Metrics**   To evaluate the effectiveness of the PAC reasoning in optimizing budget usage, we define two key metrics: *Expert Call Percentage* (ECP) and *Saved Token Percentage* (STP). These metrics are formally defined as follows:

$$\text{ECP} := \frac{|\{i : U_i \geq \hat{u}, i \in \mathcal{I}_{\text{test}}\}|}{N} \times 100\%, \quad \text{STP} := \frac{1}{N} \sum_{i \in \mathcal{I}_{\text{test}}} \frac{l_{\tilde{y}_i} + \mathbb{1}\{U_i \geq \hat{u}\}l_{y_i}}{l_{y_i}} \times 100\%, \quad (5)$$

where $l_{\tilde{y}_i}$ and $l_{y_i}$ represent the number of tokens in the candidate answer $\tilde{y}_i$ and the reference answer $y_i$, respectively. The ECP measures the proportion of test cases requiring expert intervention. At the same time, the STP quantifies the token efficiency by comparing the token counts of candidate and reference answers, accounting for cases where expert calls are triggered. We also report accuracy of our proposed method in Appendix L

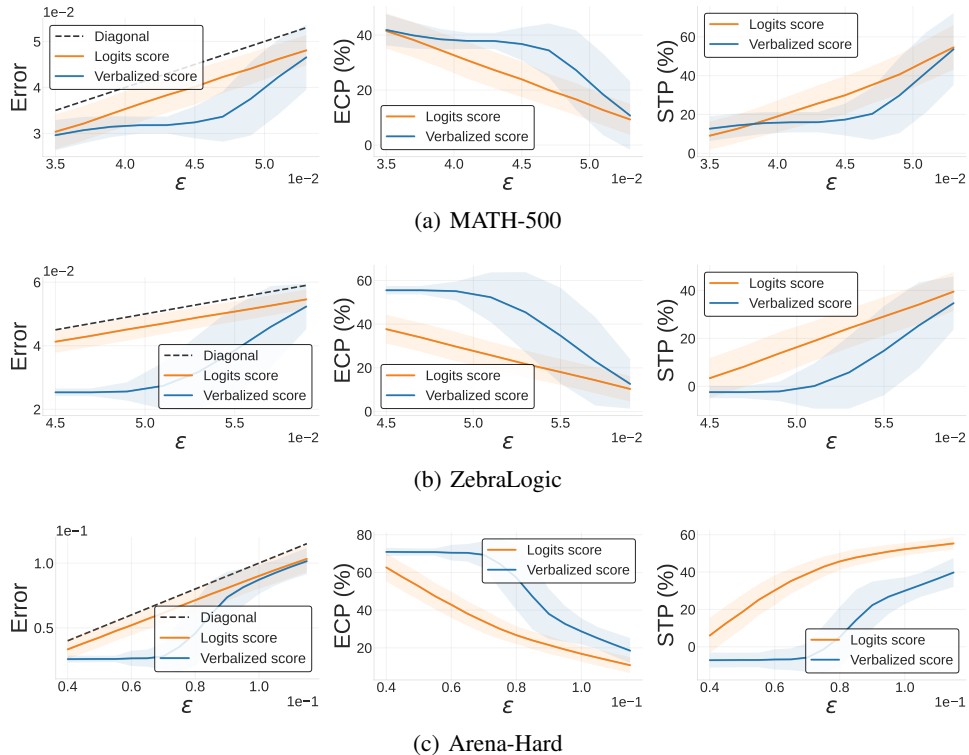

Figure 2: **Error control, ECP and STP of PAC reasoning** for semantic loss across three benchmarks at $\alpha = 0.05$. Uncertainty score includes the logits-based score and the verbalized score. All experiments are repeated 100 times, and the shaded areas represent standard deviations.

We repeat each experiment 100 times and report the mean and standard deviation of the budget savings. We fix $\alpha = 0.05$ throughout all experiments while varying $\epsilon$, and set the sampling weight $\pi = \pi_i = 0.5$ for each $i \in \mathcal{I}_{cal}$ and the sample size $m = n \times \frac{1}{\pi}$.

## 4.2 RESULTS

**PAC reasoning improves the efficiency and guarantees the performance loss.** Under the semantic loss (Figure 2), PAC reasoning consistently ensures validity of error across all the benchmarks: the empirical error remains below the target risk level $\epsilon$ while achieving substantial budget savings. For instance, on Arena-Hard with $\epsilon = 0.08$ for the logits uncertainty score, the average empirical error is approximately 0.06, the ECP is about $20\%$, and the STP is around $40\%$. Under the 0-1 binary loss (see Figure 1 and Table 3), PAC reasoning also maintains error rates within the target risk level and saves computational budget. For example, for the MATH-500 dataset, PAC reasoning saves ECP by $22.50\%$ and STP by $23.13\%$. In summary, PAC reasoning bounds the performance loss within the target risk level and significantly improves inference efficiency.

**Logits-based uncertainty score is more stable.** From the results in Figure 2 and Table 3, the logits-based uncertainty score consistently shows smoother and more stable behavior, with lower variance in both ECP and STP. In contrast, the verbalized uncertainty score exhibits larger fluctuations due to its sparse and clustered distribution. For instance, on ZebraLogic (see Table 3), the standard deviations of ECP and STP under the verbalized score are 20.68 and 17.20, considerably higher than the corresponding 7.47 and 9.90 values for the logits-based score. Similar patterns can also be observed in Figure 2, where the variance of the verbalized score is consistently higher than that of the logits-based score across different settings. Moreover, in Figure 2, the verbalized score exhibits overly conservative error control and fails to improve inference efficiency when $\epsilon$ is small. Therefore, although the verbalized score occasionally achieves stronger risk control or higher savings, its calibration is less reliable, leading to less consistent performance.

## 5  DISCUSSION

In this section we ablate the key design choices of PAC reasoning to verify its practical reliability. The ablations confirm that the framework (i) tolerates imperfect uncertainty scores, (ii) remains stable when the calibration set shrinks to only 10 % of the data, and (iii) accepts some alternative scoring functions such as off-the-shelf reward models.

**Quality of uncertainty score**    To assess the reliability of uncertainty scores, we compute the Expected Calibration Error (ECE), which measures the difference between predicted confidence and actual accuracy. Our analysis in Appendix O.1 reveals that logits-based uncertainty scores exhibit lower ECE values compared to verbalized scores on most benchmarks. For example, on MATH-500, the logits-based score achieves an ECE of 0.0450, while the verbalized score has an ECE of 0.0634. Despite these differences in calibration quality, both approaches successfully maintain the PAC guarantee, demonstrating the robustness of our framework to imperfect uncertainty quantification. This suggests that perfect calibration is not necessary for PAC reasoning to work effectively, as long as the uncertainty score provides some signal about disagreement likelihood.

**Calibration set size**    The size of the calibration set presents a practical trade-off between calibration cost and guarantee of tightness. Our comprehensive ablation study in Appendix O.2 examines calibration ratios ranging from 10% to 90% of the total dataset. The results show that error control is remarkably stable across different calibration sizes, with average loss remaining within the target tolerance even for small calibration sets. For instance, on MATH-500 with logits-based uncertainty, the average loss varies only from 0.0331 to 0.0344 as the calibration ratio increases from 10% to 90%. The STP also remains relatively stable, ranging from 14.39% to 18.52%. This demonstrates that PAC reasoning can be deployed with modest calibration costs while still providing meaningful efficiency gains and statistical guarantees.

**Alternative uncertainty scores**    Our framework is flexible and can accommodate any scoring function that correlates with the disagreement rate. An alternative is reward model-based scoring, where a trained reward model evaluates the quality of the non-thinking model's output. We conduct experiments in Appendix O.3 using reward scores on MATH-500 and ZebraLogic, demonstrating that PAC reasoning maintains valid error control even with this alternative scoring method. For example, on MATH-500 with $\epsilon = 0.045$, reward-based PAC reasoning achieves an error of 0.0362 with 22.43% token savings. This flexibility is particularly valuable in scenarios where uncertainty scores are unreliable or unavailable.

## 6  RELATED WORK

Our work intersects efficiency improvement for reasoning models and the distribution-free inference for risk control of its performance loss with confidence.

**Efficiency improvement for reasoning models**    Large Reasoning Models (LRMs) have recently become a research hotspot due to their outstanding performance in handling complex tasks (Yue et al., 2025). However, the problem of overthinking has emerged (Sui et al., 2025; Chen et al., 2025), where LRMs tend to engage in unnecessarily long reasoning chains and redundant computational steps. This increases latency and cost, and may even cause error accumulation through extended reasoning paths. For example, in mathematical problems, LRMs may explore irrelevant solution branches or perform excessive intermediate calculations that do not contribute to the final answer. To alleviate this issue, recent studies propose efficient reasoning strategies such as Early Exit of thinking (Yang et al., 2025b; Jiang et al., 2025) and adaptive switching between "fast" and "slow" thinking modes to reduce reasoning tokens and avoid redundant steps (Cheng et al., 2025; Chung et al., 2025; Fang et al., 2025; Li et al., 2025; Liang et al., 2025; Ma et al., 2025; Paliotta et al., 2025; Pan et al., 2024a;b; 2025; Xiao & Gan, 2025; Yao et al., 2024; Yong et al., 2025). Despite their empirical effectiveness, these techniques lack theoretical guarantees on the performance loss when using the non-thinking mode. We fill this gap by introducing a PAC-based reasoning that provides statistically guaranteed performance loss for efficient reasoning.

**PAC learning and Learn-then-Test framework**  PAC learning (Valiant, 1984), and the LTT framework (Bates et al., 2021; Angelopoulos et al., 2025b) provide a theoretical foundation for reliable machine learning. PAC learning establishes how algorithms can generalize from training data with probabilistic guarantees, addressing questions of sample complexity and error bounds. The LTT framework is a useful distribution-free inference method (Angelopoulos et al., 2025a; Gibbs et al., 2025; Lei & Wasserman, 2014; Lei et al., 2018), offering an approach for calibrating predictive algorithms to achieve risk control, enabling practitioners to specify error tolerances and confidence levels while constructing systems that satisfy these requirements with high probability. The key innovation of LTT is separating the learning phase from the testing phase (Zeng et al., 2025), where performance guarantees are established. This framework has been extended to conformal risk control (Angelopoulos et al., 2025c) and its applications in language modeling (Quach et al., 2024), allowing for flexible control of monotone loss functions while maintaining distribution-free guarantees. This work applies the LTT framework to reasoning model efficiency improvement and rigorously validates its theoretical properties. This application enables us to provide formal guarantees for the trade-off between computational efficiency and reasoning accuracy for LRMs.

## 7 CONCLUSION

We propose PAC reasoning, which is designed to provide rigorous, theoretically grounded, and practical efficiency improvement for reasoning models while maintaining probabilistic correctness guarantees. Our approach constructs a composite model that selectively uses either an expert model or a candidate model based on a constructed confidence bound. The PAC reasoning contributes by delivering statistical assurances for model performance and demonstrating that it reliably achieves the specified error rate with probability. We validate our method through extensive experiments on real-world reasoning tasks, showing it can significantly reduce computational costs while maintaining user-specified error tolerances with confidence. Future research will focus on developing more advanced uncertainty estimation techniques, exploring tighter theoretical bounds, and broadening the method's applicability to other large language model efficiency-improvement strategies.

**Limitations**  Though our PAC reasoning provides rigorous statistical guarantees, there are several limitations worth noting: First, the method requires access to uncertainty scores from the model without thinking, which may not always be available or reliable. Second, the sampling-based confidence bound estimation introduces additional computational overhead during calibration. Finally, the method assumes that the calibration and test distributions are i.i.d. meaning that the calibration set is drawn from the same distribution as the test set. Significant distribution shifts could invalidate the guarantees. These limitations suggest future work improving the method's robustness and applicability to a broader range of reasoning models and tasks.

**Future work**  Our current framework uses a single threshold for all inputs, which may be suboptimal when different task types or reasoning depths require different levels of model capability. We extend PAC reasoning to three promising directions. First, *conditional PAC reasoning* learns separate thresholds for different input partitions (e.g., by subject on MATH-500), with preliminary results in Appendix H showing 5-10% additional token savings while maintaining error tolerance. Second, *anytime-valid PAC reasoning* adapts to streaming data using confidence sequences (Ramdas et al., 2022; 2023), replacing fixed-sample bounds with time-uniform upper confidence bounds that remain valid under arbitrary stopping rules (Appendix I). Third, *multi-level PAC reasoning* extends beyond binary routing to multiple model tiers by learning thresholds $u_1 < u_2 < \ldots < u_k$ that route inputs based on uncertainty levels, with the PAC guarantee bounding cumulative performance loss across all tiers (Appendix J).

## REPRODUCIBILITY STATEMENT

We have made every effort to ensure that the results presented in this paper are reproducible. All code and datasets have been made publicly available in an anonymous repository an anonymous repository to facilitate replication and verification. The experimental setup, including training steps, model configurations, and hardware details, is described in detail in the paper. We have also provided a full description of the implementation of PAC reasoning, including uncertainty estimators, loss functions, and evaluation metrics, to assist others in reproducing our experiments. Additionally, we used publicly available reasoning and generation benchmarks, such as MATH-500 (Lightman et al., 2023), ZebraLogic (Lin et al., 2025), and Arena-Hard (Li et al., 2024), ensuring consistent and reproducible evaluation results. We believe these measures will enable other researchers to reproduce our work and further advance the field.

## ETHICS STATEMENT

This work adheres to the ICLR Code of Ethics. In this study, no human subjects or animal experimentation were involved. All datasets used, including MATH-500 (Lightman et al., 2023), ZebraLogic (Lin et al., 2025), and Arena-Hard (Li et al., 2024), were sourced in compliance with relevant usage guidelines, ensuring no violation of privacy. We have taken care to avoid any biases or discriminatory outcomes in our research process. No personally identifiable information was used, and no experiments were conducted that could raise privacy or security concerns. We are committed to maintaining transparency and integrity throughout the research process.

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

## A    LLM USAGE

This work was developed with the help of large language models (LLMs), including GPT-5 and Gemini 2.5 pro, which helped enhance the clarity, coherence, and technical precision of the writing. The authors' original research contributions, intellectual property, and key arguments remain entirely their own. All content has undergone thorough review and validation by the authors to ensure accuracy and scientific integrity.

## B    FROM LEARN-THEN-TEST TO PAC REASONING

The LTT framework considers a family of post-processing mappings $\{T_\lambda\}_{\lambda \in \Lambda}$. It aims to select a parameter $\hat{\lambda}$ such that the risk $R(T_{\hat{\lambda}})$ is controlled with confidence. Our PAC reasoning problem naturally fits into this framework through the following construction:

**Mapping into LTT framework**    We define the parameter space as $\Lambda = \{U_1, \ldots, U_n\}$, where $U_i$ are the uncertainty scores on the calibration set. This discretization to sample points is consistent with the implementation of our algorithm. For each threshold $u \in \Lambda$, we define the mapping:

$$T_u(x_i) := \begin{cases} y_i, & U_i \geq u \quad \text{(query the model with thinking)}, \\ \tilde{y}_i, & U_i < u \quad \text{(use the model without thinking)}. \end{cases} \tag{6}$$

This mapping $T_u$, where $u \in \Lambda$, corresponds precisely to our uncertainty-based selection strategy in the PAC reasoning framework. We define the empirical risk as the average performance loss:

$$\hat{R}(T_u) := \frac{1}{n} \sum_{i=1}^{n} \ell(y_i, T_u(x_i)) = L(u), \tag{7}$$

where $L(u)$ is our cumulative error function defined in (2). The LTT framework requires valid p-values for each parameter $\lambda$. Our confidence-bound construction in Algorithm 1 provides this. For each threshold $u$, our confidence upper bound $\hat{L}_u(\alpha)$ satisfies

$$\mathbb{P}(\mathbb{E}L(u) \leq \hat{L}_u(\alpha)) \geq 1 - \alpha.$$

This can be converted to a valid p-value for testing the null hypothesis $H_u : L(u) > \epsilon$ by defining

$$p_u := \inf\{\gamma \in (0,1) : \hat{L}_u(\gamma) \leq \epsilon\}.$$

Under the null hypothesis $H_u$, this p-value is stochastically dominated by the uniform distribution on $[0,1]$, satisfying the validity requirement of the LTT framework.

**Monotonicity and fixed-sequence testing**    A crucial property of our setup is that the risk function $u \mapsto L(u)$ is monotonically non-decreasing in $u$. This monotonicity allows us to use the fixed-sequence testing procedure from the LTT framework without additional multiple testing corrections (e.g., Benjamini-Hochberg procedure, Bonferroni correction, or Holm's step-down method). Specifically, our threshold selection $\hat{u} = \max\{u \in \Lambda : \hat{L}_u(\alpha) \leq \epsilon\}$ is equivalent to the fixed-sequence single-start procedure in LTT, which maintains the family-wise error rate control at level $\alpha$. Therefore, the theoretical guarantees of our PAC reasoning inherit the rigorous foundation of the LTT framework, ensuring that our risk control is valid despite the data-dependent threshold selection.

## C    PROOF OF THEOREM 4

*Proof.* Let
$$u^\star := \inf\{u \in \Lambda : R(u) > \epsilon\}.$$

As $R(u)$ is non-decreasing, it holds that
$$R(u) \leq \epsilon \text{ for all } u \leq u^\star,$$
and
$$R(u^\star) > \epsilon.$$

If $\hat{u} > u^\star$, then

$$\widehat{L}_{u^\star}(\alpha) \leq \epsilon < R(u^\star),$$

which contradicts Assumption 3.1 except with probability at most $\alpha$.

Therefore,

$$\mathbb{P}(\hat{u} \leq u^\star) \geq 1 - \alpha,$$

and because $R(u)$ is non-decreasing,

$$R(\hat{u}) \leq \epsilon \text{ on this event.}$$

$\square$

## D    VALIDITY OF CLT-BASED UPPER CONFIDENCE BOUND

We show that a CLT-based upper confidence bound computed on the calibration set satisfies Assumption 3.1 asymptotically.

**Proposition 7** (Asymptotic validity of UCB baed on CLT). *Fix a threshold $u$ and let $Z_j(u)$ be the i.i.d. random variables defined in Algorithm 1 for $j = 1, \ldots, m$, with mean $L(u)$ and variance $\sigma_Z^2(u) > 0$. Let $\widehat{\mu}_Z(u) = m^{-1} \sum_{j=1}^{m} Z_j(u)$ and $\widehat{\sigma}_Z^2(u) = (m-1)^{-1} \sum_{j=1}^{m} \left(Z_j(u) - \widehat{\mu}_Z(u)\right)^2$. Define the UCB based on CLT*

$$\widehat{L}_u^{\mathrm{CLT}}(\alpha) \; := \; \widehat{\mu}_Z(u) \; + \; z_{1-\alpha}\sqrt{\widehat{\sigma}_Z^2(u)/m},$$

*where $z_{1-\alpha}$ is the $(1-\alpha)$-quantile of the standard normal distribution. Then*

$$\liminf_{m \to \infty} \mathbb{P}\left(L(u) \leq \widehat{L}_u^{\mathrm{CLT}}(\alpha)\right) \; \geq \; 1 - \alpha.$$

*Proof.* By the classical Lindeberg–Feller central limit theorem,

$$\frac{\sqrt{m}\left(\widehat{\mu}_Z(u) - L(u)\right)}{\sigma_Z(u)} \to \mathcal{N}(0, 1)$$

in distribution as $m \to \infty$ because the variables are i.i.d. Since $\widehat{\sigma}_Z(u) \xrightarrow{p} \sigma_Z^2(u)$ by the weak law of large numbers, Slutsky's theorem yields

$$\frac{\sqrt{m}\left(\widehat{\mu}_Z(u) - L(u)\right)}{\sqrt{\widehat{\sigma}_Z(u)}} \to \mathcal{N}(0, 1).$$

Therefore

$$\mathbb{P}\left(\frac{L(u) - \widehat{\mu}_Z(u)}{\sqrt{\widehat{V}_Z(u)/m}} \leq z_{1-\alpha}\right) \to 1 - \alpha.$$

Equivalently, define $d_m := (L(u) - \widehat{\mu}_Z(u))/\sqrt{\widehat{V}_Z(u)/m}$ and observe that

$$\{d_m \leq z_{1-\alpha}\} = \left\{L(u) \leq \widehat{\mu}_Z(u) + z_{1-\alpha}\sqrt{\widehat{V}_Z(u)/m}\right\}.$$

Hence,

$$\liminf_{m \to \infty} \mathbb{P}\left(L(u) \leq \widehat{L}_u^{\mathrm{CLT}}(\alpha)\right) \; \geq \; 1 - \alpha.$$

$\square$

# E   PROOF OF THEOREM 5

## E.1   NOTATION RECALLING AND LEMMA

We present PAC guarantees for the empirical test risk under precise Hoeffding conditions, making explicit the roles of calibration and test randomness. For any threshold $u$, the deployment rule $T_u$ predicts with the expert when $U(x) \geq u$ and otherwise uses the fast model. The population risk is

$$R(u) \;=\; \mathbb{E}_{(x,y)\sim P}\big[\ell\big(y, T_u(x)\big)\big].$$

Given an independent test set $\mathcal{D}_{\text{test}}$ drawn i.i.d. from $P$, the empirical test risk is

$$\widehat{R}(u) \;=\; \frac{1}{N} \sum_{j=n+1}^{n+N} \ell\big(y_j, T_u(x_j)\big).$$

Our guarantee for the empirical test risk relies on Assumption 3.1 from the main text, in addition to the following lemma.

**Lemma 8** (Conditional Hoeffding bound). *Let $\hat{u}$ be a random variable determined by the calibration set $\mathcal{D}_{cal}$. Assume that, conditioned on $\hat{u}$, the test losses $Z_j(\hat{u}) := \ell\big(y_j, T_{\hat{u}}(x_j)\big)$ for $j \in \mathcal{I}_{test}$ are i.i.d. and bounded in $[a, b]$. Then for any $t > 0$,*

$$\mathbb{P}\big(\widehat{R}(\hat{u}) - R(\hat{u}) > t \,\big|\, \hat{u}\big) \;\leq\; \exp\Big(-\tfrac{2Nt^2}{(b-a)^2}\Big).$$

*Proof.* Let $Z_j(\hat{u}) = \ell\big(y_j, T_{\hat{u}}(x_j)\big)$ for $j \in \mathcal{I}_{test}$. The model $\hat{u}$ is fixed when we condition on it. Since the test set $\mathcal{D}_{\text{test}}$ consists of i.i.d. samples and is independent of $\hat{u}$, the random variables $Z_1(\hat{u}), \ldots, Z_N(\hat{u})$ are conditionally independent and identically distributed.

By the boundness of the loss function, each $Z_j(\hat{u})$ is bounded in $[a, b]$. The conditional expectation of each $Z_j(\hat{u})$ is $\mathbb{E}[Z_j(\hat{u}) \,|\, \hat{u}] = \mathbb{E}[\ell(y_j, T_{\hat{u}}(x_j)) \,|\, \hat{u}]$. Since $(x_j, y_j)$ is independent of $\hat{u}$, this is equal to the unconditional expectation over the data distribution, $\mathbb{E}_{(x,y)\sim P}[\ell(y, T_{\hat{u}}(x))]$, which is the definition of the true risk $R(\hat{u})$. The empirical risk is the sample mean:

$$\widehat{R}(\hat{u}) = \frac{1}{N} \sum_{j=1}^{N} Z_j(\hat{u}).$$

Its conditional expectation is

$$\mathbb{E}[\widehat{R}(\hat{u}) \,|\, \hat{u}] = \frac{1}{N} \sum_{j=1}^{N} \mathbb{E}[Z_j(\hat{u}) \,|\, \hat{u}] = R(\hat{u}).$$

We can now apply Hoeffding's inequality (Hoeffding, 1963) to the conditional i.i.d. bounded variables $Z_j(\hat{u})$. For any $t > 0$, the one-sided version states that

$$\mathbb{P}\left( \frac{1}{N} \sum_{j=n+1}^{n+N} Z_j(\hat{u}) - \mathbb{E}\left[ \frac{1}{N} \sum_{j=n+1}^{n+N} Z_j(\hat{u}) \,\bigg|\, \hat{u} \right] > t \,\bigg|\, \hat{u} \right) \leq \exp\left( -\frac{2Nt^2}{(b-a)^2} \right).$$

Substituting the empirical risk and true risk, we get

$$\mathbb{P}\left( \widehat{R}(\hat{u}) - R(\hat{u}) > t \,\bigg|\, \hat{u} \right) \leq \exp\left( -\frac{2Nt^2}{(b-a)^2} \right).$$

This completes the proof. $\qquad\qquad\square$

## E.2   PROOF DETAILS FOR THEOREMS 5

*Proof.* The independence of $\mathcal{D}_{\text{test}}$ and $\mathcal{D}_{cal}$ ensures Lemma 8 hold. Use the inclusion

$$\{\widehat{R}(\hat{u}) > \epsilon + t\} \subseteq \{R(\hat{u}) > \epsilon\} \cup \{\widehat{R}(\hat{u}) - R(\hat{u}) > t\}$$

and take probabilities. Combine the result of Theorem 4 (i.e., $\mathbb{P}(R(\hat{u}) > \epsilon) \leq \alpha$) with the law of total probability and Lemma 8 to conclude the claim:

$$
\begin{aligned}
\mathbb{P}(\widehat{R}(\hat{u}) > \epsilon + t) &\leq \mathbb{P}(R(\hat{u}) > \epsilon) + \mathbb{P}(\widehat{R}(\hat{u}) - R(\hat{u}) > t) \\
&= \mathbb{P}(R(\hat{u}) > \epsilon) + \mathbb{E}\big[\mathbb{P}\big(\widehat{R}(\hat{u}) - R(\hat{u}) > t \,\big|\, \hat{u}\big)\big] \\
&\leq \alpha + \exp\Big(-\frac{2Nt^2}{(b-a)^2}\Big).
\end{aligned}
$$

This is equivalent to the stated guarantee. $\qquad\square$

## F    FINITE-SAMPLE CONFIDENCE BOUNDS FOR BOUNDED LOSSES

This section presents an alternative algorithm for computing confidence bounds that provides strict finite-sample guarantees under bounded loss assumptions. This approach leverages concentration inequalities such as the Hoeffding inequality or betting-based confidence intervals (Bentkus, 2004; Hao et al., 2019; Hoeffding, 1994; Learned-Miller & Thomas, 2020; Ramdas et al., 2022; Waudby-Smith & Ramdas, 2021; 2024) to achieve tighter bounds compared to the asymptotic normal approximation used in Algorithm 1.

---

**Algorithm 3** Compute Confidence Bound $\hat{L}_u(\alpha)$ Based on Hoeffding Inequality

---

**Input:** Calibration set $\{(x_i, y_i)\}_{i=1}^n$, model with thinking $f$, model without thinking $\tilde{f}$, uncertainty scores $\{U_i\}_{i=1}^n$, sampling weights $\{\pi_i\}_{i=1}^n$, sampling size $m$, confidence level $\alpha$, loss upper bound $B > 0$

**Output:** The finite-sample upper confidence bound $\hat{L}_u(\alpha)$.

1:  Initialize an empty list of samples $\mathcal{Z} = []$.
2:  Let $\tilde{y}_i = \tilde{f}(x_i)$ for all $i = 1, \dots, n$.
3:  **for** $j = 1, \dots, m$ **do**
4:      Sample an index $i_j \sim \text{Unif}(\{1, \dots, n\})$.
5:      Sample a Bernoulli random variable $\xi_{i_j} \sim \text{Bern}(\pi_{i_j})$.
6:      **if** $\xi_{i_j} = 1$ **then**
7:          Query the true label $y_{i_j}$ and compute $Z_j = \min\Big(\frac{\ell(y_{i_j}, \tilde{y}_{i_j})}{\pi_{i_j}}, \frac{B}{\pi_{i_j}}\Big)$.
8:      **else**
9:          $Z_j = 0$.
10:     **end if**
11:     Append $Z_j$ to $\mathcal{Z}$.
12: **end for**
13: For a threshold $u$, define the variables $Z_j(u) = Z_j \cdot \mathbf{1}\{U_{i_j} \leq u\}$ for $j = 1, \dots, m$.
14: $\hat{\mu}_Z(u) \leftarrow \frac{1}{m} \sum_{j=1}^m Z_j(u)$
15: $R \leftarrow \frac{B}{\min_i \pi_i}$
16: $\delta_{\text{HB}}(\alpha) \leftarrow \sqrt{\frac{R^2 \log(2/\alpha)}{2m}}$
17: **Return** $\hat{L}_u(\alpha) \leftarrow \hat{\mu}_Z(u) + \delta_{\text{HB}}(\alpha)$.

---

## G    TRANSDUCTIVE PAC REASONING

In this section, we introduce a transductive version of PAC reasoning. In this setting, the calibration set and the test set are identical. We consider a fixed dataset $\mathcal{D} = \mathcal{D}_{test} = \mathcal{D}_{cal} = \{x_1, \dots, x_n\}$, and the randomness only comes from the algorithm itself (e.g., which data points are selected to query the expert, the sampling design, and the internal randomization of the mean upper bound estimator). The goal is to provide a guarantee of empirical performance over this fixed dataset. Specifically, the algorithm ensures that the empirical average performance loss is controlled below a user-specified tolerance level $\epsilon$ with a confidence of at least $1 - \alpha$.

Let's update the setup for the transductive setting. For a given threshold $u$, the empirical risk on the dataset $\mathcal{D}$ is defined as

$$L(u) := \frac{1}{n} \sum_{i=1}^{n} \ell(y_i, \hat{y}_i) \, \mathbf{1}\{U_i \leq u\}.$$

This represents the average loss for the data points where the model is used (i.e., uncertainty is below the threshold $u$), and $L_u$ is non-decreasing in $u$. The validity of our transductive PAC reasoning algorithm relies on the following assumptions. For any given threshold $u$ and significance level $\alpha$, there exists an upper confidence bound (UCB) $\widehat{L}_u(\alpha)$, computable from samples drawn by the algorithm, that satisfies

$$\mathbb{P}\big(L(u) \leq \widehat{L}_u(\alpha)\big) \geq 1 - \alpha.$$

In practice, we instantiate $\widehat{L}_u(\cdot)$ using our UCB procedures. Concretely, one may compute $\widehat{L}_u(\alpha')$ via Algorithm 1 (CLT-based) or Algorithm 3 (Hoeffding/Bentkus/betting-based), depending on sample size and desired conservatives. We summarize our transductive style method in Algorithm 4.

---

**Algorithm 4** Transductive PAC-Labeling

1: **Inputs:** Test ataset $\mathcal{D} = \{x_1, \ldots, x_n\}$, uncertainty scores $U_i, i = 1, \ldots n$, model predictions $\tilde{y}_i, \tilde{y}_n$, tolerance $\epsilon$, significance level $\alpha$, number of trials $m$, sampling probabilities $\pi_i, i = 1, \ldots n$, UCB function $\hat{L}_u(\alpha)$.
2: **Output:** Labeled dataset $\{(X_i, \widetilde{Y}_i)\}_{i=1}^{n}$ and threshold $\hat{u}$.
3: *Sampling phase:*
4: **for** $j = 1, \ldots, m$ **do**
5:     Draw an index $i_j$ according to the sampling design (e.g., uniform or importance-based).
6:     With probability $\pi_{i_j}$, query the expert for $y_{i_j}$. Let $\xi_j \sim \text{Bernoulli}(\pi_{i_j})$.
7:     **if** $\xi_j = 1$ **then**
8:         Observe the true label $y_{i_j}$ and compute the loss $\ell(y_{i_j}, \tilde{y}_{i_j})$.
9:     **else**
10:         Mark as not-observed.
11:     **end if**
12: **end for**
13: *For each candidate threshold, compute UCB $\hat{L}_u(\alpha)$*
14: *Choose the estimated threshold $\hat{u} := \max\{\widehat{L}_u \leq \varepsilon\}$.*
15: **for** $i = 1, \ldots, n$ **do**
16:     **if** $U_i \geq \hat{u}$ **then**
17:         Ensure expert label $y_i$ is obtained (query now if not already queried).
18:         Set $\widetilde{y}_i := y_i$.
19:     **else**
20:         Set $\widetilde{y}_i := \hat{y}_i$.
21:     **end if**
22: **end for**

---

**Theorem 9** (Transductive PAC guarantee). *Suppose $\mathbb{P}\big(L(u) \leq \widehat{L}_u(\alpha)\big) \geq 1 - \alpha$. Then the procedure in Algorithm 4, which selects $\hat{u} = \max\{u : \widehat{L}_u(\alpha) \leq \varepsilon\}$, achieves*

$$\mathbb{P}\big(L(\hat{u}) \leq \varepsilon\big) \geq 1 - \alpha.$$

*Proof.* Let $E$ be the event that $L_u \leq \widehat{L}_u(\alpha)$ holds simultaneously for all $u \in \Lambda$. Then $\mathbb{P}(E) \geq 1 - \alpha$. On $E$, for any $u$ with $\widehat{L}_u(\alpha) \leq \varepsilon$ we have $L(u) \leq \varepsilon$. By the selection rule, either the set is non-empty and $\hat{u}$ is the maximal such $u$, which implies $L_{\hat{u}} \leq \varepsilon$ by monotonicity, and we take $\hat{u} = -\infty$, in which case $L_{\hat{u}} = 0 \leq \varepsilon$ since no model predictions are used. Therefore $\mathbf{1}\{L(\hat{u}) \leq \varepsilon\} = 1$ on $E$, and hence $\mathbb{P}(L_{\hat{u}} \leq \varepsilon) \geq \mathbb{P}(E) \geq 1 - \alpha$. $\qquad\square$

*Remark* 10 (Comparison with the inductive setting). In the inductive setting, the calibration and test sets are drawn independently of the same distribution. The goal is to guarantee performance on future, unseen data points from that distribution. The guarantee is of the form $\mathbb{P}(\mathbb{E}L(\hat{u}) \leq \varepsilon) \geq 1 - \alpha$, where the probability is over the random draws of both datasets. In contrast, our

transductive approach provides a guarantee for a specific, fixed dataset, which can be more suitable in applications where the set of items to be labeled is known in advance or the calibration and test datasets are not exchangeable.

# H CONDITIONAL PAC REASONING

In this part, we extend PAC reasoning to the conditional setting on the MATH-500 dataset. Each problem in MATH-500 is annotated with a subject category, which allows us to estimate a separate threshold $\hat{u}$ for every subject group. We adapt our PAC reasoning conditionally on the subject. All experiments are repeated 100 times. The results are presented in Figure 3. The standard (marginal) PAC reasoning fails to maintain valid risk control across individual subject groups, even though it may control the overall error. In contrast, **the proposed conditional PAC reasoning successfully achieves risk validity both for each subject group and for the total population.** The cost of this stronger, conditional guarantee is a small degradation in ECP performance, as illustrated in Figure 3.

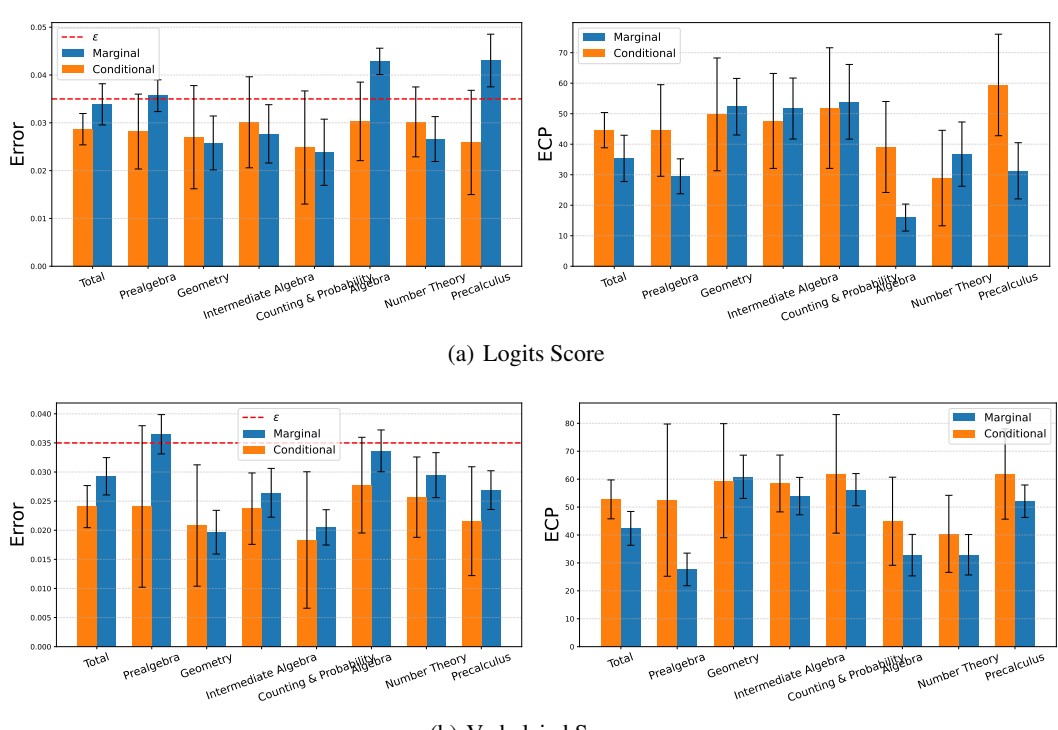

(a) Logits Score

(b) Verbalzied Score

Figure 3: Error control and ECP of **conditional** PAC reasoning for semantic loss on the MATH-500 dataset at a confidence level of $\alpha = 0.05$ and a risk level $\epsilon = 0.04$.

# I ANYTIME VALID PAC REASONING

In sequential deployment, we require risk control that is valid at all times and under arbitrary stopping rules. Under a stable distribution with no drift, we use time-uniform upper bounds to control risk and adapt the uncertainty threshold over time. Let $R(u) = \mathbb{E}[\ell(y, T_u(x))]$ denote the population risk parameterized by the uncertainty threshold $u$, and let $\widehat{R}_t(u)$ be its empirical counterpart built from the first $t$ observations. A confidence sequence is a family of time-uniform intervals $\{CS_t(u; \alpha)\}_{t \geq 1}$ such that $\mathbb{P}\left(\forall t \geq 1 : R(u) \in CS_t(u; \alpha)\right) \geq 1 - \alpha$. We instantiate the upper bound via a modular constructor, yielding a time-uniform upper bound $\text{UCB}_t(u; \alpha)$ under choices such as Hoeffding or Bernstein, and choose the largest threshold whose UCB remains below the loss tolerance $\epsilon$.

**Drift-aware anytime valid**  When drift may occur, we adopt a drift-aware anytime-valid PAC reasoning scheme based on an e-process. We construct nonnegative e-values from uncertainty scores and optionally losses, and accumulate them as a test martingale. Let $L_t(u) = \ell(y_t, \tilde{y}_t)\,\mathbf{1}\{U_t \le u\}$. Using the source window, estimate an empirical CDF $\hat{F}_{\mathrm{src}}$ of uncertainty score and define a sequential p-value $p_t = \hat{F}_{\mathrm{src}}(U_t)$. With a calibration parameter $\lambda \in (0,1)$, transform to an e-value via the power e-function $e_t = p_t^{\lambda-1}$, which satisfies $\mathbb{E}_{\mathrm{src}}[e_t \mid \mathcal{H}_{t-1}] \le 1$ under no drift. Form an e-process $M_t = \prod_{i=T_0+1}^{t} e_i$. By Ville's inequality, a reset is triggered when $M_t \ge 1/\beta$.

**Confidence sequence**  We assigns a time-varying confidence budget $\alpha_t$ satisfying $\sum_{t=1}^{\infty} \alpha_t \le \alpha$ to ensure validity at all times. Following designs in Jamieson et al. (2014) and Howard et al. (2021), the resulting anytime-valid upper bound is

$$\mathrm{UCB}_t(\alpha') = \hat{\mu}_t + \sqrt{\frac{B^2 \,\log\!\left(\frac{C\,t^{1.1}}{\alpha'}\right)}{2t}}\,.$$

Here $\hat{\mu}_t = t^{-1}\sum_{i=1}^{t} L_i$ is the empirical mean of historical losses. The constant $B$ upper bounds each loss, for instance $B = 1$ for the 0–1 loss. This bound grows mildly faster than $\log(\log t)$, trading tightness for strict anytime validity across all sample sizes.

## J  MULTI LEVEL PAC REASONING

We extend PAC reasoning to a three-tier composite LRM while preserving the statistical guarantees. We introduce two nonthinking LRMs, denoted by $\tilde{f}_1$ and $\tilde{f}_2$, whose computational costs and accuracies lie below the model with thinking $f$. For each input prompt $x$, the expert output is $y = f(x)$, and the nonthinking outputs are $\tilde{y}_1 = \tilde{f}_1(x)$ and $\tilde{y}_2 = \tilde{f}_2(x)$. We assume the existence of a unified uncertainty score $U(x) \in [0,1]$ produced inexpensively, for example by $\tilde{f}_1$, as in Section 3. We define the deployment mapping with two thresholds $u_1, u_2 \in [0,1]$ and $u_1 \le u_2$:

$$T_{u_1,u_2}(x) = f(x)\,\mathbf{1}\{U(x) \ge u_2\} + \tilde{f}_2(x)\,\mathbf{1}\{u_1 < U(x) \le u_2\} + \tilde{f}_1(x)\,\mathbf{1}\{U(x) \le u_1\}.$$

The composite LRM is $\hat{f} = T_{\hat{u}_1,\hat{u}_2}$ for calibrated thresholds $(\hat{u}_1, \hat{u}_2)$. We re-parameterize the population risk and empirical risk of the composite LRM by the threshold pair:

$$R(u_1,u_2) = \mathbb{E}[\ell(y, T_{u_1,u_2}(x))], \quad \widehat{R}(u_1,u_2) = \frac{1}{N}\sum_{i \in \mathcal{I}_{test}} \ell(y_i, T_{u_1,u_2}(x_i)).$$

Expanding the loss conditional on the tiers yields

$$R(u_1,u_2) = \mathbb{E}\big[\ell(y,\tilde{f}_1(x))\,\mathbf{1}\{U \le u_1\} + \ell(y,\tilde{f}_2(x))\,\mathbf{1}\{u_1 < U \le u_2\}\big],$$

because the tier $U > u_2$ uses the expert $f$ and contributes zero loss. The risk function is monotone in both coordinates. Fixing $u_2$, increasing $u_1$ assigns more inputs to the cheaper $\tilde{f}_1$, which weakly increases the risk. Fixing $u_1$, increasing $u_2$ defers less often to $f$ and assigns more inputs to $\tilde{f}_2$, which also weakly increases the risk. This bi-variate monotonicity enables valid fixed-sequence calibration and preserves the PAC guarantee as in Definition 1. We construct a two-dimensional upper confidence bound $\widehat{L}_{u_1,u_2}(\alpha)$ satisfying

$$\mathbb{P}\big(R(u_1,u_2) \le \widehat{L}_{u_1,u_2}(\alpha)\big) \ge 1 - \alpha \quad \text{for all valid pairs } (u_1,u_2) \in [0,1]^2,\ u_1 \le u_2.$$

The UCB can be built on the calibration set via importance sampling, using partial expert queries with sampling probabilities $\pi_i$ and weights, analogously to Algorithms 1 and 3. Specifically, when an expert label $y_i$ is queried, we record two weighted losses $Z_{i,1} = \ell(y_i, \tilde{y}_{i,1})/\pi_i$ and $Z_{i,2} = \ell(y_i, \tilde{y}_{i,2})/\pi_i$, otherwise we record zeros, and then aggregate tier-wise according to $(u_1, u_2)$. Under a central limit theorem or concentration inequalities, we obtain a valid $\widehat{L}_{u_1,u_2}(\alpha)$. Once the UCB is available, we calibrate the thresholds by searching over the empirical grid of unique uncertainty scores in the calibration set and selecting a pair that minimizes computation under the constraint $\widehat{L}_{u_1,u_2}(\alpha) \le \epsilon$. A simple and effective choice is to maximize $u_2$ subject to validity and then break ties by maximizing $u_1$, which prioritizes using $\tilde{f}_2$ and $\tilde{f}_1$ more often while respecting the

Table 1: Prompt for the verbalized confidence scores.

---

**System prompt:** You are a reasoning assistant. For each question and proposed answer, you must estimate how likely the proposed answer is correct.

**User prompt:**

Question: {QUESTION}

Answer: {ANSWER}

Provide a probability (between 0.0 and 1.0) that your answer is correct. Only output the probability.

---

target tolerance. By monotonicity and the validity of $\widehat{L}_{u_1,u_2}(\alpha)$, the selected pair $(\hat{u}_1, \hat{u}_2)$ ensures $R(\hat{u}_1, \hat{u}_2) \leq \epsilon$ with probability at least $1 - \alpha$. The multi-level extension therefore preserves the PAC efficiency improvement while enabling finer control over computation across multiple tiers.

**Generalization to K-tier PAC reasoning**  This framework extends directly to K-tier systems with $K - 1$ nonthinking LRMs ordered by cost and accuracy. Let $\tilde{f}_1, \ldots, \tilde{f}_{K-1}$ be the nonthinking LRMs and introduce thresholds $0 \leq u_1 \leq \cdots \leq u_{K-1} \leq 1$. Define the deployment mapping for a threshold vector $\mathbf{u} = (u_1, \ldots, u_{K-1})$ as

$$T_{\mathbf{u}}(x) = \tilde{f}_1(x)\,\mathbf{1}\{U(x) \leq u_1\} + \sum_{k=2}^{K-1} \tilde{f}_k(x)\,\mathbf{1}\{u_{k-1} < U(x) \leq u_k\} + f(x)\,\mathbf{1}\{U(x) > u_{K-1}\}.$$

The population risk is

$$R(\mathbf{u}) = \mathbb{E}\Big[\ell\big(y, \tilde{f}_1(x)\big)\,\mathbf{1}\{U \leq u_1\} + \sum_{k=2}^{K-1} \ell\big(y, \tilde{f}_k(x)\big)\,\mathbf{1}\{u_{k-1} < U \leq u_k\}\Big],$$

which is coordinate-wise non-decreasing in each threshold $u_k$. A valid upper confidence bound $\widehat{L}_{\mathbf{u}}(\alpha)$ is constructed by recording $K - 1$ weighted losses when querying the expert and aggregating tier-wise. Threshold calibration proceeds on the empirical grid to minimize computation under the constraint $\widehat{L}_{\mathbf{u}}(\alpha) \leq \epsilon$, and the fixed-sequence strategy applies under monotonicity. The multi-tier deployment then uses $T_{\mathbf{u}}$ on the test set.

## K  EXPERIMENTAL DETAILS

**Hyperparameter settings of LLMs**  In this study, we configure the decoding parameters as follows: for Qwen/Qwen3-4B-Instruct-2507, we set *Temperature* = 0.7, *TopP* = 0.8, *TopK* = 20, and *MinP* = 0; for Qwen/Qwen3-4B-Thinking-2507, we set *Temperature* = 0.6, *TopP* = 0.95, *TopK* = 20, and *MinP* = 0. Experiments were run on one NVIDIA RTX A6000 Graphics Card.

**The prompt for verbalized uncertainty score**  In Table 1, we present the prompt used to elicit the verbalized confidence scores. After ten trials, we obtained the average confidence score and defined the verbalized uncertainty score as 1 minus this average confidence.

**Details of Datasets**  Table 2 summarizes the datasets employed in our experiments, together with their corresponding splitting strategies. For each dataset, we report its type, overall size, and the partitioning into PAC calibration and PAC test sets.

**Loss function**  We consider two types of loss functions for evaluating the PAC guarantee of our reasoning method. The first is a semantic cosine distance, which measures the semantic similarity between outputs in the embedding space. Formally, given reference output $y_i = f(x_i)$ and PAC reasoning output $\hat{y}i = \hat{f}(x_i)$, we compute their embeddings $v_{y_i}$ and $v_{\hat{y}_i}$, and define the loss as:

$$\ell(y_i, \hat{y}_i) = 1 - \frac{v_y \cdot v_{\hat{y}}}{\|v_y\|\|v_{\hat{y}}\|}. \tag{8}$$

Table 2: The details of datasets and splitting settings for PAC experiments

| Dataset | Dataset Type | Dataset Size | Split Setting | Size |
|---------|-------------|-------------|--------------|------|
| MATH-500 | Math Reasoning | 500 | PAC Calibration | 300 |
| | | | PAC Test | 200 |
| ZebraLogic | Text reasoning | 1000 | PAC Calibration | 500 |
| | | | PAC Test | 500 |
| Arena-Hard | Alignment Task | 750 | PAC Calibration | 450 |
| | | | PAC Test | 300 |

For the semantic embedding model, we adopt "Qwen3-Embedding-4B" (Yang et al., 2025a). Secondly, we employ a binary 0–1 loss, which captures the actual loss in answer accuracy when comparing the PAC reasoning output $\hat{y}$ with the reference output $y$:

$$\ell(y_i, \hat{y}_i) = \ell(y_i, \hat{y}_i | y_{i,gold}) = \mathbb{1}\{\hat{y}_i \neq y_i^{gold}\}\mathbb{1}\{y_i = y_i^{gold}\} \tag{9}$$

where $y_{i,gold}$ is the ground-truth answer for the problem $x_i$.

**The choice of loss functions**    The loss functions, shown as in Eq. (8) and Eq. (9), serve distinct purposes in evaluating the PAC reasoning. The semantic cosine distance captures the degree of semantic alignment between the PAC reasoning's prediction and reference outputs. It is particularly suitable for tasks where nuanced differences in meaning are critical, such as natural language understanding or generation tasks. By leveraging the "Qwen/Qwen3-Embedding4B" model, we ensure that the embeddings capture rich contextual information, robustly comparing semantic content in high-dimensional spaces. In contrast, the binary 0–1 loss is designed for scenarios where the correctness of the generated answer is verifiable, such as in mathematical problem-solving or multiple-choice question answering. This loss function is particularly effective for evaluating the framework's ability to produce exact matches to ground-truth answers, emphasizing precision in verifiable tasks. By testing the PAC reasoning on these two loss functions, we can assess the semantic quality and factual accuracy of the PAC reasoning across diverse tasks.

## L    ACCURACY OF PAC REASONING BASED ON SEMANTIC LOSS

In this section, we investigate the effectiveness of the PAC reasoning framework when controlled by semantic loss. As shown in Figure 4, applying semantic loss to regulate the PAC filtering process leads to consistently improved final accuracies across both MATH-500 and ZebraLogic. Both logits-based and verbalized uncertainty scores yield higher Pass@1 performance than the baseline nonthinking model, with the verbalized score performing the best. The results indicate that PAC reasoning controlled by semantic loss reliably enhances output accuracy while remaining robust to different $\epsilon$ settings.

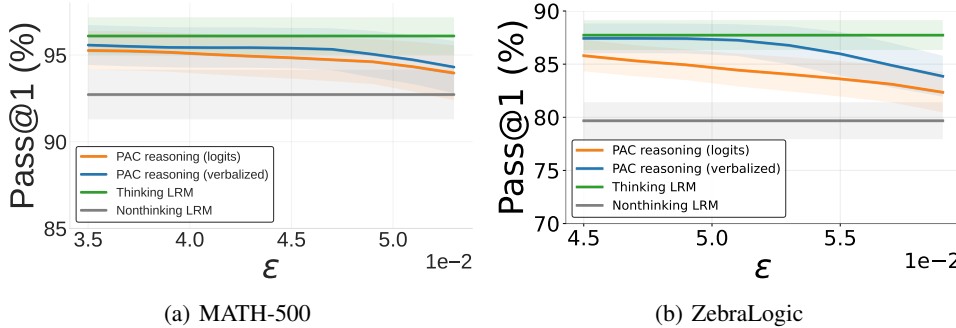

(a) MATH-500                                                    (b) ZebraLogic

Figure 4: Accuracy of PAC reasoning based on semantic loss across mathematical benchmarks at a confidence level of $\alpha = 0.05$. Uncertainty score includes the logits-based score and the verbalized score. All experiments are repeated 100 times, and the shaded areas represent standard deviations.

## M  MAIN RESULTS FOR THE BINARY LOSS

For the binary loss, we evaluate PAC reasoning on the verifiable datasets MATH-500 and Ze-braLogic, with target risk levels set to $\epsilon = 0.03$ and $\epsilon = 0.08$, respectively (see Section 4.1 for experimental details). For comparison, we also consider a naive fixed-threshold baseline as well as the approach that relies solely on the nonthinking model.

As shown in Table 3, PAC reasoning consistently keeps the error rates below the target risk, while also achieving substantial efficiency gains. In contrast, the naive baseline exhibits unstable behavior across datasets: on ZebraLogic, although it attains a very small error with logits-based uncertainty, it violates efficiency by yielding a negative STP ($-34.78\%$), meaning it requires even more tokens than fully using the thinking model. Meanwhile, on MATH-500 with verbalized uncertainty, the same method produces a large error ($0.0346$), which substantially exceeds the target risk $\epsilon = 0.03$. These results highlight that naive threshold fails to provide reliable control over both loss and budget, often swinging between overly conservative and overly risky outcomes. In summary, PAC reasoning strikes a balanced trade-off, keeping the error within $\epsilon$ while delivering consistent savings across tasks and datasets.

Table 3: Experimental results of the binary loss function on verifiable datasets ($\alpha = 0.05$). For MATH-500, we set $\epsilon = 0.03$, and for ZebraLogic, we set $\epsilon = 0.08$.

| Dataset | Metric | Logits-based score | | Verbalized score | | Non-thinking |
|---|---|---|---|---|---|---|
| | | PAC reasoning | Naive ($U_i \geq 0.05$) | PAC reasoning | Naive ($U_i \geq 0.05$) | |
| MATH-500 | Error | $0.0206 \pm 0.0126$ | $0.0179 \pm 0.0068$ | $0.0209 \pm 0.0141$ | $0.0346 \pm 0.0095$ | $0.0435 \pm 0.0107$ |
| | ECP (%) ↓ | $21.48 \pm 17.85$ | $14.44 \pm 2.02$ | $24.59 \pm 20.48$ | $2.83 \pm 0.94$ | – |
| | STP (%) ↑ | $37.61 \pm 23.19$ | $43.58 \pm 4.78$ | $36.13 \pm 26.44$ | $66.67 \pm 4.91$ | – |
| ZebraLogic | Error | $0.0615 \pm 0.0181$ | $0.0062 \pm 0.0026$ | $0.0530 \pm 0.0246$ | $0.0631 \pm 0.0074$ | $0.1163 \pm 0.0102$ |
| | ECP (%) ↓ | $22.50 \pm 7.47$ | $77.28 \pm 1.36$ | $26.95 \pm 20.68$ | $12.49 \pm 1.07$ | – |
| | STP (%) ↑ | $23.13 \pm 9.90$ | $-34.78 \pm 1.26$ | $21.21 \pm 17.20$ | $32.70 \pm 2.11$ | – |

## N  EXPERIMENTAL RESULTS OF MORE LLMS

In this section, we evaluate PAC reasoning on additional LLM architectures and larger-scale models to further verify the generalizability of our framework. Specifically, we conduct experiments using Llama-3.1-8B–based models: the "DeepSeek-R1-Distill-Llama-8B" as the thinking model and "Llama-3.1-8B-Instruct" as the lower-performance nonthinking model. We configure the decoding parameters as follows: for Llama-3.1-8B-Instruct, we set *Temperature* = 0.6, *TopP* = 0.95, *TopK* = 20, and *MinP* = 0, *max_tokens* = 4096; for DeepSeek-R1-Distill-Llama-8B, we set *Temperature* = 0.6, *TopP* = 0.95, *TopK* = 20, and *MinP* = 0. Experiments were run on one NVIDIA RTX A6000 Graphics Card. Other experimental details are following Appendix K.

Figure 5 summarizes the performance of PAC reasoning on Llama-3.1-8B–based models. Across all three benchmarks, PAC reasoning consistently maintains valid error control, with empirical errors staying below the diagonal reference line. For uncertainty estimation, the logits-based score exhibits tighter calibration and lower ECP than the verbalized score, particularly under smaller $\epsilon$ values, while the verbalized score shows slightly higher variance but still adheres to theoretical bounds. In terms of efficiency, both scores achieve substantial STP, demonstrating that PAC reasoning can reliably identify confident cases and reduce unnecessary calls to the thinking model. Overall, the results confirm that PAC reasoning generalizes well to larger LLM architectures and continues to deliver stable risk control and efficiency gains.

## O  EXTENSIVE STUDY

### O.1  EXPECTED CALIBRATION ERROR OF TWO UNCERTAINTY SCORES

We evaluate the calibration quality of uncertainty estimates on MATH-500 and ZebraLogic using Qwen3-4B-Instruct-2507. We consider two uncertainty scores: a logits-based score derived from

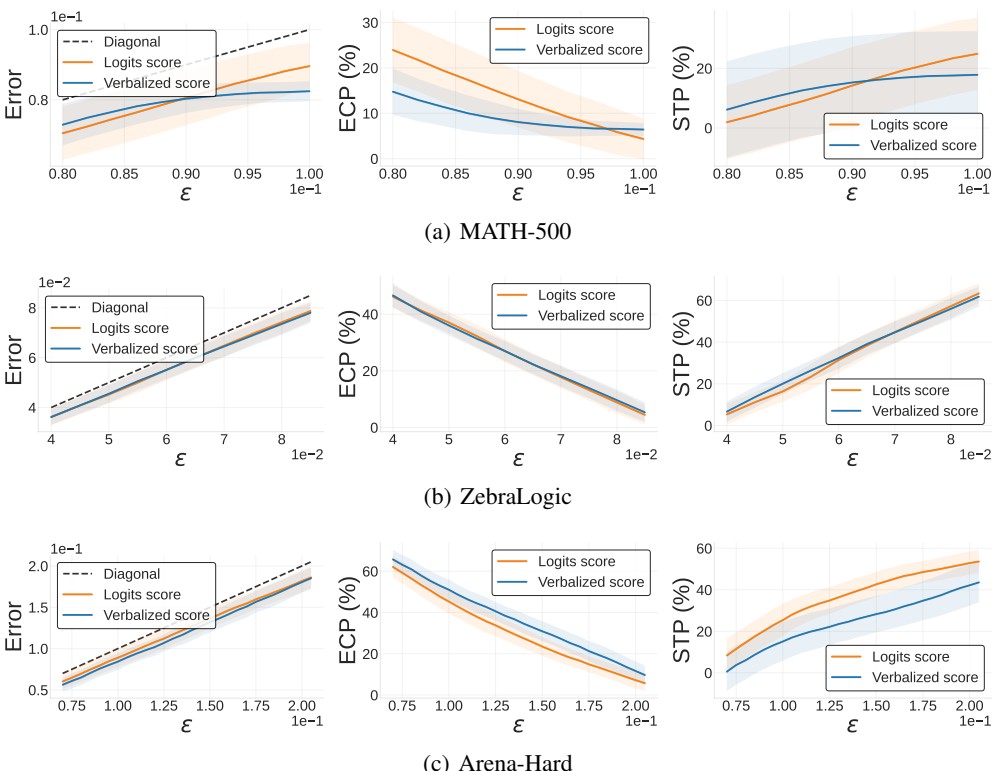

Figure 5: Error control, ECP and STP of PAC reasoning for semantic loss across three benchmarks under a confidence level of $\alpha = 0.05$, on the Llama-3.1-8B–based models. Uncertainty score includes the logits-based score and the verbalized score. All experiments are repeated 100 times, and the shaded areas represent standard deviations.

the model's predictive distribution and a verbalized score elicited from the model's self-reported confidence. Expected calibration error (ECE) (Guo et al., 2017) quantifies the discrepancy between predicted confidence and empirical accuracy via binning and a weighted average of absolute gaps, where smaller values indicate better calibration. Across both benchmarks, the logits-based score exhibits consistently lower ECE and smoother reliability than the verbalized score, indicating tighter calibration of uncertainty estimates. The verbalized score shows higher variance and mild overconfidence in high-confidence bins. These findings support the use of the logits-based score within PAC reasoning and motivate improved elicitation methods for verbalized confidence. Figure 6 summarizes the reliability plots and aggregated ECE. All experiments are repeated 100 times under the decoding configuration described in Section 4.1.

## O.2 ABLATION STUDY ON THE SIZE OF THE CALIBRATION SET

We conduct experiments to investigate the stability of efficiency gains under different correction set sizes. Specifically, we repeat the experiments with varying calibration ratios to examine how the size of the correction set influences performance. The results are presented in Figure 7. PAC reasoning maintains stable error control and consistent uncertainty calibration across all benchmarks. Both uncertainty scoring methods are capable of controlling the theoretical risk, though the verbalized score exhibits larger variance. Moreover, the logits-based score consistently outperforms the verbalized uncertainty score, achieving lower ECP and higher STP. These findings demonstrate that our framework, i.e., PAC reasoning, can effectively maintain valid risk control and stable efficiency gains under varying calibration dataset sizes.

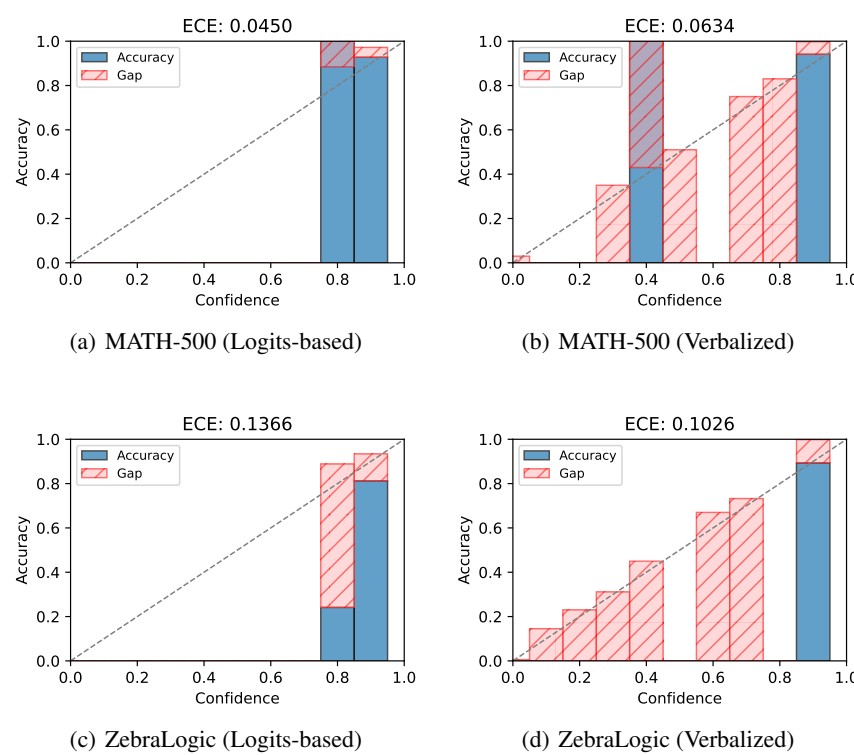

(a) MATH-500 (Logits-based)      (b) MATH-500 (Verbalized)

(c) ZebraLogic (Logits-based)      (d) ZebraLogic (Verbalized)

Figure 6: Expected Calibration Error across two mathematical benchmarks.

### O.3   REWARD SCORE AS AN ALTERNATIVE UNCERTAINTY SCORE

In this part, we evaluate whether PAC reasoning remains valid when replacing the original uncertainty score with the reward score. Concretely, we apply PAC reasoning to MATH-500 and ZebraLogic using the reward score as the uncertainty estimate, and we report its error control based on the semantic cosine distance, ECP, and STP under varying $\epsilon$. We follow the experimental setting described in Section 4.1 and the reward model is "Qwen2.5-Math-PRM-7B" (Zhang et al., 2025).

The results are presented in Figure 8. Across both benchmarks, the observed error curves remain below the diagonal baseline, indicating that PAC reasoning still satisfies the theoretical error guarantee even with this alternative scoring method. For efficiency, ECP consistently decreases as $\epsilon$ increases, showing that the method becomes more selective. These results show that PAC reasoning is robust to the choice of uncertainty score: using the reward score still ensures valid error control and provides reasonable efficiency gains.

### O.4   EXPERIMENTS ON MORE DATASETS

In this section, we further validate the effectiveness of our framework on additional datasets, including GPQA (Rein et al., 2024) and HumanEval (Chen et al., 2021). The experimental results shown in Figure 9 evaluate the performance of PAC reasoning on these benchmarks using two types of uncertainty scores: the logits-based score and the verbalized score. Across both datasets, the two uncertainty scores demonstrate valid error control. The ECP increases steadily as the tolerance level $\epsilon$ grows, with the verbalized score exhibiting slightly worse ECP performance. For STP, the logits-based score consistently achieves higher performance compared to the verbalized score. Overall, these observations indicate that PAC reasoning remains effective and reliable when applied to a broader range of datasets.

### O.5   COMPARISON WITH MORE METHODS

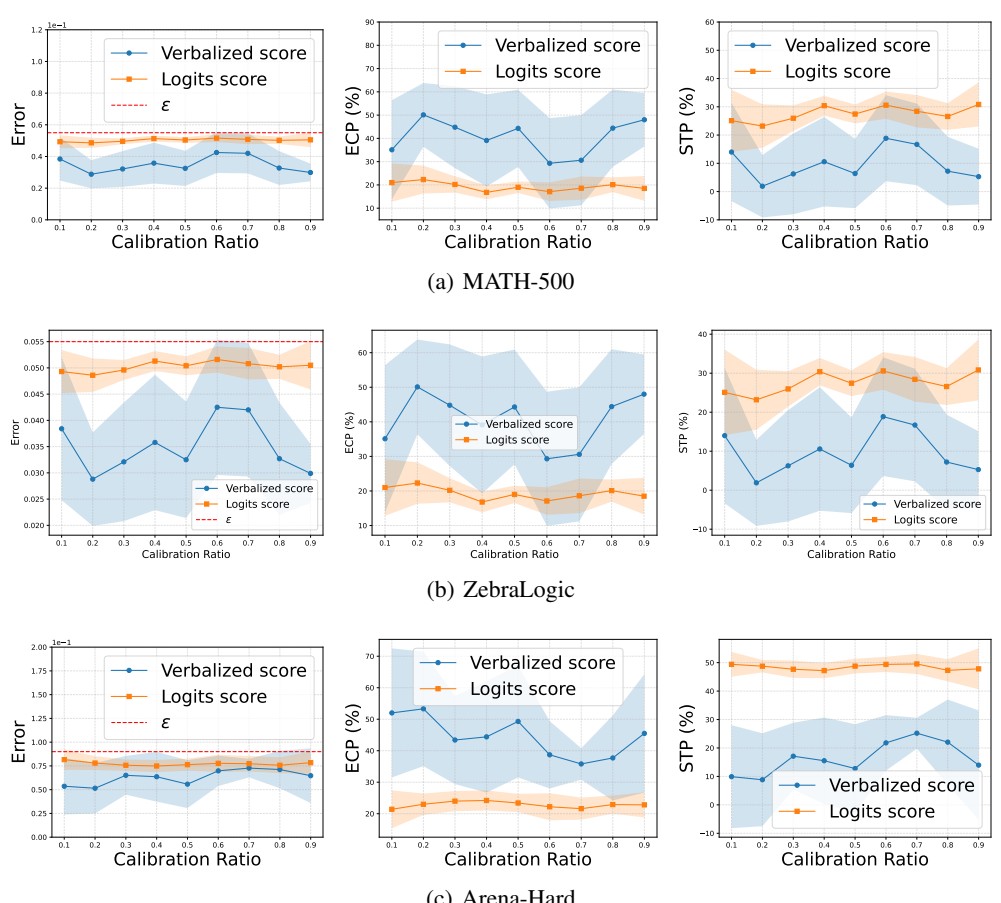

(a) MATH-500

(b) ZebraLogic

(c) Arena-Hard

Figure 7: Error control, ECP and STP of PAC reasoning for semantic loss for different calibration ratios at a confidence level of $\alpha = 0.05$. Uncertainty score includes the logits-based score and the verbalized score. The red dashed line $\epsilon$ means the target risk level, and the shaded areas represent standard deviations. All experiments are repeated 100 times.

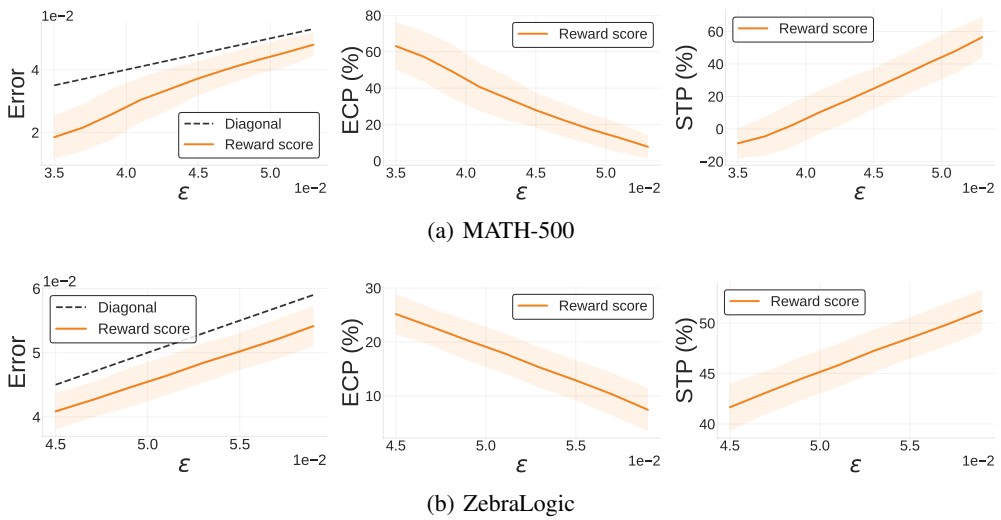

(a) MATH-500

(b) ZebraLogic

Figure 8: Error control, ECP and STP of PAC reasoning for semantic loss across two mathematical benchmarks at a confidence level of $\alpha = 0.05$. Uncertainty score is the reward score. All experiments are repeated 100 times, and the shaded areas represent standard deviations.

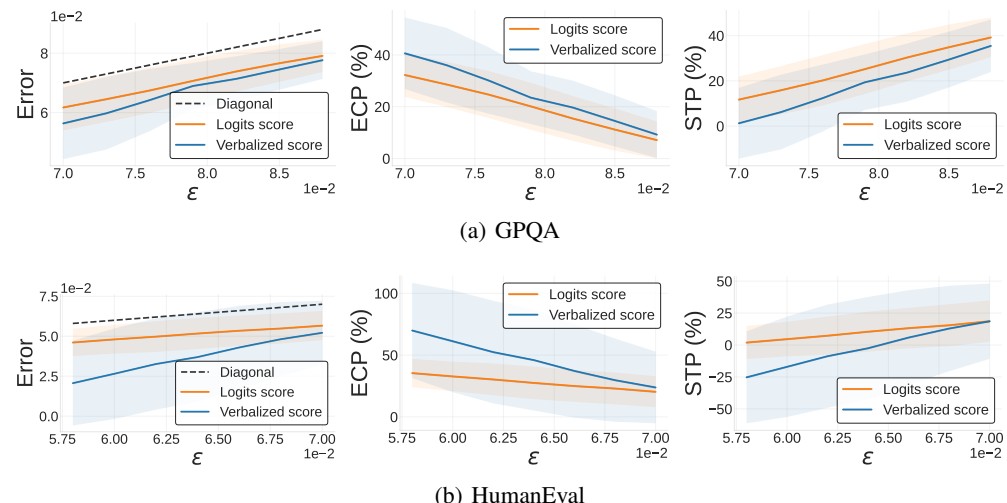

Figure 9: Error control, ECP and STP of PAC reasoning for semantic loss for GPQA and HumanEval at a confidence level of $\alpha = 0.05$. Uncertainty score includes the logits-based score and the verbalized score. The red dashed line $\epsilon$ means the target risk level, and the shaded areas represent standard deviations. All experiments are repeated 100 times.

In this section, we further compare PAC reasoning with more efficient reasoning methods, including Chain of Draft (CoD) (Xu et al., 2025) and NoThinking (Ma et al., 2025). We also include a **naive control** baseline. Specifically, given an error target $\epsilon$, it tunes a maximum threshold $u$ such that the loss $L(u) = \frac{1}{n} \sum_{i=1}^{n} \ell(y_i, \tilde{y}_i) \mathbf{1}\{U_i \leq u\} < \epsilon$ on the calibration set. Because this approach ignores the uncertainty in estimating $L(u)$, it lacks the inductive guarantees provided by our PAC-based reasoning method. We conduct the experiments on the MATH-500 dataset based on binary 0-1 loss, following the experimental setting in Section K. We choose the logits-based score to quantify generation uncertainty.

Table 4 shows that PAC Reasoning is the only method that satisfies the target error tolerance, benefiting from its theoretical guarantees. Unlike heuristic approaches such as CoD and NoThinking, which exhibit large and uncontrolled errors, PAC reasoning maintains the lowest loss while achieving substantial efficiency gains. This demonstrates that our method enables safe and effective inference-cost reduction, clearly distinguishing it from existing heuristic approaches.

Table 4: Experimental results of different methods under $\epsilon = 0.03$.

| Metric | Naive control | PAC reasoning | CoD | NoThinking |
|---|---|---|---|---|
| Binary Loss | $0.0351 \pm 0.0094$ | $0.0206 \pm 0.0126$ | $0.3548 \pm 0.0604$ | $0.4445 \pm 0.0583$ |
| STP (%) ↑ | $61.53 \pm 6.06$ | $37.61 \pm 23.19$ | $-1.33 \pm 0.82$ | $-4.51 \pm 1.00$ |
| Pass@1 | $0.93 \pm 0.25$ | $0.95 \pm 0.23$ | $0.61 \pm 0.49$ | $0.50 \pm 0.50$ |

