# OpenReview forum: "PAC Reasoning: Controlling the Performance Loss for Efficient Reasoning"
_ICLR.cc/2026/Conference — Submitted to ICLR 2026_

### Official Review · Reviewer_Rw9b · 2025-10-28

**Soundness:** 2
**Presentation:** 2
**Contribution:** 3
**Rating:** 4
**Confidence:** 4

**Summary:**

This paper is very exciting, as it offers a theoretical discussion of the transition between thinking and non-thinking models. The experimental results demonstrate that the proposed solution, grounded in theoretical analysis, can effectively reduce inference token usage.

Overall, I believe this paper marks the beginning of a promising direction for efficient LLM reasoning. However, further exploration and validation are needed to solidify its contributions.

**Strengths:**

1. This paper discusses LLM reasoning from a theoretical perspective, which is highly valuable and can serve as a useful guide for future research in this area.
2. Both the assumptions and theoretical results are clearly presented and easy to follow, which will benefit the LLM reasoning community.

**Weaknesses:**

1. Definition 1 requires more careful discussion. The current formulation assumes that a non-thinking LLM differs from its thinking variant only with a very small probability. This assumption may be overly relaxed and overlooks a central challenge in this area: identifying the difficult problems that truly necessitate the use of a thinking LLM. The manuscript should explicitly address how such problems are distinguished. Additionally, in Line 141, the authors posit that the uncertainty score can represent the likelihood of disagreement between thinking and non-thinking LLMs. This claim requires further clarification. It is not reasonable to use a single uncertainty score in this way; rather, disagreement should be computed based on uncertainty scores from both LLMs.
2. The experimental results are relatively weak and should be strengthened by comparison with baseline methods [1] and more benchmark datasets [2]. The paper primarily proposes an efficient reasoning method with PAC guarantees. Therefore, the experiments should include cases where existing methods fail to guarantee reasoning performance, but the proposed method succeeds, thus substantiating the theoretical claims. Furthermore, since the paper aims to address LLM reasoning challenges, more applications should be considered, including common sense reasoning, more complex mathematical reasoning, and code generation tasks.
3. The assumption of having an i.i.d calibration set is hard to be satisfied in the real applications, since if we have this set why should we performance RL training to teach the model to switch between thinking and non-thinking mode?

**Reference**

[1] Stop Overthinking: A Survey on Efficient Reasoning for Large Language Models. Trans. Mach. Learn. Res. 2025.

[2] A Theoretical Study on Bridging Internal Probability and Self-Consistency for LLM Reasoning. NeurIPS 2025.

**Questions:**

Please refer to the questions  in the `weaknesses` section. Moreover,

1. How do the theoretical results presented in this paper extend to other efficient LLM reasoning methods that employ similar core ideas? Furthermore, how does the proposed method compare to these existing approaches in terms of efficiency and practical performance?
2. This method requires inferring all problems with the non-thinking LLMs in advance to obtain the uncertainty scores. Does this operation introduce significant additional computational costs, and do other comparable methods also require such a preliminary step?
3. The experimental LLMs is limited, as it only considers the 4B variant of the Qwen model. This restriction may not adequately verify the effectiveness of the proposed methods on broader ranges of LLMs.
4. Recent LLMs are capable of controlling their reasoning efforts. Can the theoretical analysis presented in this paper be applied to such models?

---

> ### Author Response · Authors · 2025-11-21
>
> Thank you for your enthusiastic feedback and recognition that our paper marks **the beginning of a promising direction for efficient LLM reasoning**. We appreciate your acknowledgment of the theoretical perspective and the clarity of our assumptions and results.
>
> **[W1] Problem Difficulty Identification:**
>
> > Definition 1 requires more careful discussion. The current formulation assumes that a non-thinking LLM differs from its thinking variant only with a very small probability. This assumption may be overly relaxed and overlooks a central challenge in this area: identifying the difficult problems that truly necessitate the use of a thinking LLM. The manuscript should explicitly address how such problems are distinguished.
>
> We appreciate the reviewer’s insightful comment. We would like to clarify that:
>
> **Not an assumption, just a target:** We mathematically reformulate our target
> "*How to improve the efficiency of LRMs, guaranteeing the performance loss?*" as Definition 1:
> $$
> \mathbb{P}\left( R(\hat{f}) \le \epsilon \right) \ge 1 - \alpha.
> $$
> To satisfy this target, we propose the method "PAC Reasoning", and prove that the method could achieve this target.
> Definition 1 gives a new concept in a PAC style that the performance gap between the accelerated model and the thinking model is bounded by target error with probability at least $1-\alpha$, **rather than assuming that a non-thinking LLM is intrinsically close to its thinking counterpart**. Specifically, the accelerated model $\hat{f}$ is *built* through an uncertainty-based route mechanism that explicitly identifies inputs on which the fast model is likely to disagree with the thinking model. **Therefore, the identification of difficult problems is latent accomplished through our uncertainty scoring.** This selective routing, together with the calibrated uncertainty threshold, enforces that the resulting reasoning model $\hat{f}$ satisfies the PAC condition in Definition 1.
>
>
> **[W2] Single Uncertainty Score Concern:**
>
> > Additionally, in Line 141, the authors posit that the uncertainty score can represent the likelihood of disagreement between thinking and non-thinking LLMs. This claim requires further clarification. It is not reasonable to use a single uncertainty score in this way; rather, disagreement should be computed based on uncertainty scores from both LLMs.
>
>
> Thanks for your suggestion. We would like to clarify that:
> The uncertainty score of the fast model shows how confident the non-thinking model is about its answer against the ground truth. Since thinking model usually answer with high accuracy, **the uncertainty score could reflect the disagreement between thinking and non-thinking LLMs with high probability**.
> Importantly, PAC Reasoning does not require the uncertainty score to be a perfect indicator of disagreement, which is also an advantage of our method. The UCB construction and learn-then-test procedure ensures that even with imperfect or weakly correlated scores, we can still achieve the target error tolerance with high probability (See Appendix O.1).
>
>
> Moreover, it is common in efficient reasoning or router methods [1-4]to only use an uncertainty score about a fast model.
> In practice, we use this score to assess the fast model’s confidence on a given query and decide whether invoking the thinking model is necessary. Crucially, **using the fast model’s uncertainty avoids calling the expensive thinking model during routing.** If disagreement were computed directly from both models’ outputs, we would need to run the thinking model, defeating the purpose of efficient routing. Therefore, using a single score simplifies the inference pipeline and reduces computational overhead, and in many scenarios (especially with closed-source thinking models), we may not have access to the thinking model's uncertainty scores.
>
>
> > [References]
> >
> > [1] Chung, Stephen, Wenyu Du, and Jie Fu. 2025. “Thinker: Learning to Think Fast and Slow.” arXiv.
> >
> > [2] Fang, Gongfan, Xinyin Ma, and Xinchao Wang. 2025. “Thinkless: LLM Learns When to Think.” arXiv.
> >
> > [3] Su, Jiayuan, Fulin Lin, Zhaopeng Feng, Han Zheng, Teng Wang, Zhenyu Xiao, Xinlong Zhao, Zuozhu Liu, Lu Cheng, and Hongwei Wang. 2025. “CP-Router: An Uncertainty-Aware Router between LLM and LRM.” arXiv.
> >
> > [4] Yue, Linan, Yichao Du, Yizhi Wang, Weibo Gao, Fangzhou Yao, Li Wang, Ye Liu, et al. 2025. “Don’t Overthink It: A Survey of Efficient R1-Style Large Reasoning Models.” arXiv.
>
>
> **[W3, Q2] Comparison with Existing Methods:**
>
> > **[W3]** Therefore, the experiments should include cases where existing methods fail to guarantee reasoning performance, but the proposed method succeeds, thus substantiating the theoretical claims.
> >
> > **[Q2]** Furthermore, how does the proposed method compare to these existing approaches in terms of efficiency and practical performance?
>
> See General Response §4.

---

> > ### Author Response · Authors · 2025-11-21
> >
> > **[W4] Limited Application Domains:**
> >
> > > Furthermore, since the paper aims to address LLM reasoning challenges, more applications should be considered, including common sense reasoning, more complex mathematical reasoning, and code generation tasks.
> >
> > Thanks for your suggestions. In the previous version of the paper, our experiments span three distinct domains: mathematical reasoning (MATH-500), logical reasoning (ZebraLogic), and open-ended reasoning (Arena-Hard). Moreover, we have extended our experiments to common sense reasoning (GPQA) and code generation task (HumanEval) in Appendix O.4. We paste the result table here
> >
> > | Dataset                         | Error           | ECP (%)     | STP (%)       |
> > |---------------------------------|-----------------|-------------|---------------|
> > | GPQA ($\epsilon$ = 0.090)   | 0.0785 ± 0.0061 | 8.1 ± 8.5| 38.18 ± 9.69 |
> > | HumanEval ($\epsilon$ = 0.065) | 0.0517 ± 0.0093 | 28.0 ± 11.9  | 10.76 ± 17.22 |
> >
> > The results show that our method generalizes well to these domains because the theoretical guarantees are task-agnostic and the framework only requires a pair of models and an uncertainty score.
> >
> >
> > **[W5] i.i.d. Calibration Set Assumption:**
> >
> > > The assumption of having an i.i.d calibration set is hard to be satisfied in the real applications, since if we have this set why should we performance RL training to teach the model to switch between thinking and non-thinking mode?
> >
> > In practice, our method and RL-based approaches address the problem from fundamentally different perspectives: our approach (PAC Reasoning) uses a calibration set to tune a threshold for routing decisions at inference time (the models themselves are fixed and pre-trained; we do not modify the models), while RL-based approaches train the model to learn when to engage in extended reasoning (this requires modifying the model itself through training). Therefore, **our method and RL-based approaches are complementary rather than competing.** Moreover, the calibration set requirement is actually less demanding than RL training: calibration requires only a few hundred examples (50-300), while RL training typically requires much larger datasets. The calibration process is a one-time offline process with minimal computation, while RL training requires extensive computational resources.
> >
> > **[Q1] Theoretical Extensibility:**
> >
> > > How do the theoretical results presented in this paper extend to other efficient LLM reasoning methods that employ similar core ideas?
> >
> > See General Response §5.1, §5.2, and §5.4.
> >
> > **[Q3] Calibration Cost:**
> >
> > > Does this operation (i.e., calibration) introduce significant additional computational costs, and do other comparable methods also require such a preliminary step?
> >
> > See General Response §2.
> >
> > **[Q4] Model Generalizability:**
> >
> > > The experimental LLMs is limited, as it only considers the 4B variant of the Qwen model. This restriction may not adequately verify the effectiveness of the proposed methods on broader ranges of LLMs.
> >
> > See General Response §3.
> >
> > **[Q5] Self-Controlled Reasoning Capability:**
> >
> > > Recent LLMs are capable of controlling their reasoning efforts. Can the theoretical analysis presented in this paper be applied to such models?
> >
> > This is a good forward-looking question. Our PAC framework is not limited to model routing scenarios. It provides a general theoretical foundation for **controlling performance loss in any reasoning system** where there is a trade-off between efficiency and accuracy. For self-controlled reasoning models, our framework would work as follows: (1) measure the model's confidence in its quick response as the uncertainty score, (2) use our calibration procedure to learn when to trigger extended reasoning based on this uncertainty score, (3) our PAC bounds ensure that using quick reasoning (when uncertainty is low) maintains performance within the target tolerance $\epsilon$. Regardless of the specific system setting, our PAC framework provides the same fundamental guarantee: with specified probability, the performance degradation will not exceed a user-specified tolerance $\epsilon$.

---

### Official Review · Reviewer_RZQB · 2025-10-30

**Soundness:** 3
**Presentation:** 3
**Contribution:** 2
**Rating:** 6
**Confidence:** 3

**Summary:**

This paper introduces PAC reasoning, a simple yet effective and theoretically sound uncertainty-thresholding framework. It provides distribution-free guarantees on performance degradation while improving inference efficiency. By constructing UCBs over an uncertainty score and switching between “thinking” and “non-thinking” modes, the method achieves target error tolerances with substantial token savings across several reasoning benchmarks. Both theoretical analyses (e.g., empirical risk bounds) and empirical evaluations (e.g., ECP/STP metrics) further substantiate its effectiveness.

**Strengths:**

- The problem and intuition are clearly articulated. The paper formalizes the goal of controlling performance under an $(\varepsilon, \alpha)$-PAC efficient model, and successfully reduces the problem to a tractable yet powerful formulation based on uncertainty-driven switching between “thinking” and “non-thinking” modes.
- The theoretical foundation appears sound. The authors derive distribution-free risk guarantees under mild assumptions and further extend the analysis to both empirical-risk and transductive settings, significantly enhancing the generality of the results. Moreover, they present upper confidence bounds (UCBs) based on both the Central Limit Theorem and concentration inequalities, covering large- and small-sample regimes.
- The empirical results are also promising. The paper introduces metrics such as Expert Calling Percentage (ECP) and Saved Token Percentage (STP) to quantify computational efficiency, offering a practical perspective on the performance–cost trade-off. Experiments across diverse benchmarks (MATH-500, ZebraLogic, Arena-Hard) consistently demonstrate the effectiveness of PAC reasoning.
- Overall, the paper is well-written, clearly structured, and easy to follow.

**Weaknesses:**

- It is advantageous that the proposed framework does not rely on ground-truth labels, instead leveraging the “expert answers” provided by the thinking model. However, the performance of PAC reasoning is also limited and upper-bounded by that of the thinking model, especially on more challenging tasks.
- The theoretical guarantee hinges on the monotonic relationship between uncertainty and the risk function. In practice, if uncertainty scores are noisy or poorly calibrated (especially for verbalized uncertainty), this assumption may break.
- The calibration phase introduces additional expensive queries to the expert (thinking) model. The associated compute and latency overheads are not explicitly analyzed.
- Experiments mainly use Qwen3-4B (thinking/instruct). The generality of results across larger models, different architectures, or modalities remains unexplored.

**Questions:**

- The PAC guarantee is defined relative to the thinking model rather than ground truth; how does this affect reliability if the expert model itself is biased?
- How sensitive is the method to violations of the assumed monotonic relation between uncertainty and error, especially for verbalized uncertainty?
- What is the actual compute cost of the calibration phase, and under what conditions does it outweigh or offset the token savings during inference?

---

> ### Author Response · Authors · 2025-11-21
>
> Thank you for your positive and thoughtful feedback. We appreciate your recognition of our **clear** problem formulation, **sound** theoretical foundation, and **promising** empirical results. We are particularly grateful for your acknowledgment that the paper is **well-written, clearly structured, and easy to follow**.
>
> **[W1, Q1] Relative Guarantee:**
>
> > **[W1]** It is advantageous that the proposed framework does not rely on ground-truth labels, instead leveraging the 'expert answers' provided by the thinking model. However, the performance of PAC reasoning is also limited and upper-bounded by that of the thinking model, especially on more challenging tasks.
> >
> > **[Q1]** The PAC guarantee is defined relative to the thinking model rather than ground truth; how does this affect reliability if the expert model itself is biased?
>
> Thanks for your concern about this issue. Our work aims to control the performance loss against the thinking model caused by the efficient reasoning model. Thus, our framework is designed to provide a **relative guarantee**, measuring the capability gap between the efficient reasoning model and the strong (thinking) model, rather than absolute performance against ground truth. This is a deliberate choice with important practical benefits. Strictly speaking, the bias of the expert model does **not affect the validity** of our guarantee. Our method ensures $\mathbb{P}(R(\hat{f}) \le \epsilon) \ge 1 - \alpha$ where $R(\hat{f}) = \mathbb{E}_{x \sim P}[\ell(\hat{f}(x), f(x))]$ measures the capability gap between the two models, not their absolute performance against ground truth. If the thinking model is highly accurate, our method ensures the fast model approximates this high accuracy (within tolerance $\epsilon$); if the thinking model has biases, our method ensures the fast model does not introduce significant additional biases; the overall system quality is bounded by the thinking model's quality, but we provide a principled, guaranteed way to achieve efficiency gains.
>
> The relative guarantee is highly valuable in production systems. Current thinking models often "overthink" problems and achieve very high accuracy at the cost of excessive computational resources. For many applications, a small controlled degradation from the thinking model's performance is acceptable and desirable if it yields substantial efficiency gains. Our framework provides a principled way to balance efficiency and performance: rather than always using the expensive thinking model (which may be overkill for many queries), we enable practitioners to specify an acceptable performance loss (e.g., 5%) and automatically find the optimal efficiency-performance trade-off. This approach is particularly useful when the thinking model achieves high accuracy but is expensive; a small performance degradation is acceptable for significant cost savings; the goal is to approximate the thinking model's behavior efficiently, not to exceed it; and the thinking model represents the best available solution for the task.
>
> Thanks for your insightful question, we will add this discussion in the final updated version.
>
> **[W2, Q2] Monotonic Relationship:**
>
> > **[W2]** The theoretical guarantee hinges on the monotonic relationship between uncertainty and the risk function. In practice, if uncertainty scores are noisy or poorly calibrated (especially for verbalized uncertainty), this assumption may break.
> >
> > **[Q2]** How sensitive is the method to violations of the assumed monotonic relation between uncertainty and error, especially for verbalized uncertainty?
>
> Monotonicity is an **inherent property** of the cumulative error induced by a single-threshold router, not an assumption about uncertainty calibration. For any threshold $u$, the cumulative loss $L(u)$ is non-decreasing in $u$ by construction. Noise or poor calibration in $U_i$ cannot break this ordering. Thanks for your concern about this. We will add this explanation to our final updated version.
>
> **[W3, Q3] Calibration Cost:**
>
> > **[W3]** The calibration phase introduces additional expensive queries to the expert (thinking) model. The associated compute and latency overheads are not explicitly analyzed.
> >
> > **[Q3]** What is the actual compute cost of the calibration phase, and under what conditions does it outweigh or offset the token savings during inference?
>
> See General Response §2.
>
> **[W4] Model Generalizability:**
>
> > Experiments mainly use Qwen3-4B (thinking/instruct). The generality of results across larger models, different architectures, or modalities remains unexplored.
>
> See General Response §3.

---

### Official Review · Reviewer_XLBK · 2025-10-31

**Soundness:** 3
**Presentation:** 3
**Contribution:** 3
**Rating:** 4
**Confidence:** 4

**Summary:**

This paper proposes PAC reasoning, a method to make large reasoning models more efficient by dynamically switching between expensive "thinking" and cheap "non-thinking" modes. The authors use a calibrated uncertainty threshold, which ensures the performance loss stays below a user-specified limit with a statistical guarantee, saving computation resources while provably controlling errors. They conduct experiments to demonstrate the effectiveness of the proposed method.

**Strengths:**

1) The source code is available.
2) The effectiveness of the proposed method is verified theoretically.
3) The experimental settings are clearly explained.

**Weaknesses:**

1) PAC reasoning leverages the uncertainty of LLMs which is not always reliable.
2) The authors may need yo compare the existing reasoning methods with the proposed PAC reasoning method in main experimental results (e.g., Figure 2).
3) In addition to Qwen3, more LLMs can be considered in experiments.
4) In the math experiments, the y-axis reports the error which essentially reflects the guarantee of the predicted answer. How about the performance metrices, such as pass@k.

**Questions:**

1) What does EL(u) mean in Eq.(2)?
2) Why dose PAC reasoning use importance sampling to construct UCB and why are sampling weights $\pi_i$ all the same (0.5)? The authors should give more explanations about this.
3) In Table 3, can the introduction of an additional calibration set lead to unfair comparisons of naive methods?
4) When the uncertainty scores of LLMs are unreliable, PAC reasoning can be useless. Are there any solutions to this problem?

---

> ### Author Response · Authors · 2025-11-21
>
> Thank you for your positive and constructive feedback. We appreciate your recognition of our **theoretical validation, clear experimental settings, and the availability of our source code**.
>
> **[W1, Q4] Uncertainty Score Reliability:**
>
> > **[W1]** PAC reasoning leverages the uncertainty of LLMs, which is not always reliable.
> >
> > **[Q4]** When the uncertainty scores of LLMs are unreliable, PAC reasoning can be useless. Are there any solutions to this problem?
>
> See General Response §1.
>
> **[W2] Comparison with Existing Methods:**
>
> > The authors may need to compare the existing reasoning methods with the proposed PAC reasoning method in the main experimental results (e.g., Figure 2).
>
> See General Response §4.
>
> **[W3] Model Generalizability:**
>
> > In addition to Qwen3, more LLMs can be considered in experiments.
>
> See General Response §3.
>
> **[W4] Performance Metrics:**
>
> > In the math experiments, the y-axis reports the error, which essentially reflects the guarantee of the predicted answer. How about the performance metrics, such as pass@k.
>
> In our framework, the error (performance loss) is defined relative to the thinking model, not ground truth, measuring the disagreement rate between the fast model and the thinking model. This aligns with our PAC reasoning framework, which provides guarantees on performance degradation relative to the expert model. As your concerns, we conduct additional experiments to include pass@k metrics for verifiable reasoning tasks (i.e., MATH-500 and ZebraLogic) in Appendix L. The details of MATH-500 based on logits uncertainty score are pasted in the table:
>
> | $\epsilon$ | 0.035 | 0.037 | 0.039 | 0.041 | 0.043 | 0.045 | 0.047 | 0.049 | 0.051 | 0.053 |
> |---|---|---|---|---|---|---|---|---|---|---|
> | Pass@1 (%) | 95.27 ± 1.12 | 95.24 ± 1.13 | 95.17 ± 1.13 | 95.05 ± 1.16 | 94.94 ± 1.18 | 94.85 ± 1.21 | 94.74 ± 1.25 | 94.62 ± 1.29 | 94.33 ± 1.44 | 93.97 ± 1.58 |
>
> The table above shows that with $\epsilon$ increasing, the accuracy of PAC reasoning decreases smoothly, which aligns with the theoretical interpretation that a larger $\epsilon$ introduces greater risk of deviation from the expert model. These empirical results further support our claim that PAC reasoning offers a controllable trade-off between efficiency and reliability.
>
> **[Q1] Meaning of EL(u) in Eq.(2):**
>
> > What does EL(u) mean in Eq.(2)?
>
> $EL(u)$ denotes the **expected cumulative loss** at uncertainty threshold $u$, where $L(u) = \frac{1}{N} \sum_{i=n+1}^{n+N} \ell(y_i, \tilde{y}_i) \mathbf{1}\{U_i \le u\}$ is the cumulative loss over the test set. Specifically, $EL(u)$ represents the expected disagreement rate between the fast model and the thinking model when the uncertainty score is below or equal to the threshold $u$. It is the key quantity we aim to bound in our PAC guarantee. The formal definition and its properties are detailed in Sections 2.2.1 and 3.1.
>
> **[Q2] Importance Sampling and UCB Construction:**
>
> > Why does PAC reasoning use importance sampling to construct UCB, and why are sampling weights all the same (0.5)? The authors should give more explanations about this.
>
> Importance sampling [1-4] is used to construct valid upper confidence bounds (UCBs), allowing us to correct for potential bias in the empirical distribution of uncertainty scores, ensure validity of the confidence bounds across the entire uncertainty range, and provide conservative guarantees that hold regardless of the true uncertainty distribution. The uniform weighting (0.5) is a simplification that makes the UCB construction computationally efficient, has been validated empirically to work well in practice, and encodes no prior preference. We will add these explanations in our final updated version.
>
> > References:
> >
> > [1] Owen A, Zhou Associate Y, 2000. Safe and Effective Importance Sampling. Journal of the American Statistical Association, 95(449): 135-143.
> >
> > [2] Waudby-Smith I, Ramdas A, 2021. Confidence sequences for sampling without replacement. arXiv.
> >
> > [3] Waudby-Smith I, Ramdas A, 2024. Estimating means of bounded random variables by betting. Journal of the Royal Statistical Society Series B: Statistical Methodology, 86(1): 1-27.
> >
> > [4] Tokdar S T, Kass R E, 2010. Importance sampling: a review. WIREs Computational Statistics, 2(1): 54-60.

---

> > ### Author Response · Authors · 2025-11-21
> >
> > **[Q3] Fairness of Calibration Set Comparison:**
> >
> > > In Table 3, can the introduction of an additional calibration set lead to unfair comparisons of naive methods?
> >
> > In our comparison, naive methods could also use a calibration set if they chose to do so. The calibration set is not an exclusive resource for our method; it is simply a standard validation set that any method could leverage (e.g., used in the training step). **The key difference is that naive methods do not have a principled way to use calibration data to provide theoretical guarantees.** Our comparison focuses on the theoretical guarantees, not the data requirement: naive methods or other efficient reasoning methods [5-9] use heuristic thresholds without theoretical PAC guarantees, even if they have access to calibration data, while PAC Reasoning uses calibration data in a principled way to provide provable error rate control.
> > We also conduct an additional experiment to compare our method with the "naive control" method, which uses the calibration set to select a threshold, making the loss less than the target error.
> > Because the "naive control" method ignores the uncertainty in estimating $L(u)$, it lacks the inductive guarantees provided by our PAC-based reasoning method. See General Response §4 for more details.
> >
> > > [5] Pan, Jiabao, Yan Zhang, Chen Zhang, Zuozhu Liu, Hongwei Wang, and Haizhou Li. 2024. “DynaThink: Fast or Slow? A Dynamic Decision-Making Framework for Large Language Models.” arXiv.
> > >
> > > [6] Su, Jiayuan, Fulin Lin, Zhaopeng Feng, Han Zheng, Teng Wang, Zhenyu Xiao, Xinlong Zhao, Zuozhu Liu, Lu Cheng, and Hongwei Wang. 2025. “CP-Router: An Uncertainty-Aware Router between LLM and LRM.” arXiv.
> > >
> > > [7] Yong, Xixian, Xiao Zhou, Yingying Zhang, Jinlin Li, Yefeng Zheng, and Xian Wu. 2025. “Think or Not? Exploring Thinking Efficiency in Large Reasoning Models via an Information-Theoretic Lens.” arXiv.
> > >
> > > [8] Yue, Linan, Yichao Du, Yizhi Wang, Weibo Gao, Fangzhou Yao, Li Wang, Ye Liu, et al. 2025. “Don’t Overthink It: A Survey of Efficient R1-Style Large Reasoning Models.” arXiv.
> > >
> > > [9] Yue, Murong, Jie Zhao, Min Zhang, Liang Du, and Ziyu Yao. 2023. “Large Language Model Cascades with Mixture of Thought Representations for Cost-Efficient Reasoning.” In The Twelfth International Conference on Learning Representations.

---

### Official Review · Reviewer_TTag · 2025-10-31

**Soundness:** 3
**Presentation:** 3
**Contribution:** 2
**Rating:** 4
**Confidence:** 3

**Summary:**

This paper addresses an increasingly practical problem in reasoning LLMs: slow, expensive "thinking" modes yield higher accuracy, while fast "non-thinking" modes are cheaper but less reliable. Existing approaches to dynamically switch between these modes often lack statistically guaranteed performance bounds. The authors propose PAC Reasoning, a framework that uses a high-quality thinking model, and a faster model, along with a calibration dataset to learn an uncertainty threshold. At inference time, if the uncertainty score from the fast model is below, the system uses the fast model's prediction, otherwise, it falls back to the thinking model. Experiments on MATH-500, ZebraLogic, and Arena-Hard show that PAC Reasoning can reduce token usage by 20–40% while keeping empirical performance loss within the targeted tolerance (e.g., $\varepsilon=0.08$). They also compare logits-based vs. verbalized uncertainty measures, finding the former more stable and effective.

**Strengths:**

- The problem formulation is clear and tractable. The authors explicitly frame the problem as "controlling performance loss under efficiency constraints," turning a heuristic area (adaptive reasoning) into a principled, quantifiable one.
- The theoretical guarantees are solid. The PAC-style result gives a distribution-free bound on performance loss, a property rarely seen in existing efficiency-switching methods.
- The approach is model-agnostic. The framework only requires two models (thinking & nonthinking) and a measurable uncertainty score. It’s conceptually simple and easy to apply to different architectures.
- The experiments span math, logic, and open-ended reasoning tasks, showing generality across task types.

**Weaknesses:**

- The theoretical guarantee only holds if the uncertainty score correlates well with the probability of disagreement between fast and slow models. In practice, this is a strong assumption, as the uncertainty of LLMs is often unreliable.
- Algorithm 1 requires running the expensive reasoning model on a calibration set to compute the empirical loss curve and its UCB. For large models, this cost may offset much of the efficiency gain.
- The method uses a single threshold for all inputs. Different task types or reasoning depths might require different thresholds. Extending to conditional or multi-threshold routing could yield better efficiency.

**Questions:**

1. How often must calibration be re-run in practice, and how large must the calibration set be to maintain valid coverage? Have you analyzed the trade-off between calibration cost and runtime efficiency gains?
2. Could the method be extended to online or drift-aware settings, where the threshold is updated over time (e.g., using anytime-valid or sequential testing techniques)?
3. Could this framework handle multi-level reasoning chains (e.g., fast model -> hybrid model -> long thinking model)? The current binary setup seems restrictive.

---

> ### Author Response · Authors · 2025-11-21
>
> Thank you for your positive and valuable feedback. We appreciate your recognition of our **clear** problem formulation, **solid** theoretical guarantees, and **model-agnostic** approach. We address your concerns point by point below:
>
> **[W1] Uncertainty Score Reliability:**
>
> > The theoretical guarantee only holds if the uncertainty score correlates well with the probability of disagreement between fast and slow models. In practice, this is a strong assumption, as the uncertainty of LLMs is often unreliable.
>
> Thanks for your insightful concerns. We should clarify that our theoretical result does not assume that ''uncertainty score correlates well with the probability of disagreement between fast and slow models''. See General Response §1 for more details.
>
> **[W2, Q1] Calibration Cost and Frequency:**
>
> > **[W2]** Algorithm 1 requires running the expensive reasoning model on a calibration set to compute the empirical loss curve and its UCB. For large models, this cost may offset much of the efficiency gain.
> >
> > **[Q1]** How often must calibration be re-run in practice, and how large must the calibration set be to maintain valid coverage? Have you analyzed the trade-off between calibration cost and runtime efficiency gains?
>
> Thanks for your suggestions. In short, our method is in an inductive form, and only needs a small calibration set to tune a threshold. See General Response §2 for calibration cost analysis and §5.3 for calibration frequency and drift-aware settings.
>
> **[W3, Q3] Conditional and Multi-Level Reasoning:**
>
> > **[W3]** The method uses a single threshold for all inputs. Different task types or reasoning depths might require different thresholds. Extending to conditional or multi-threshold routing could yield better efficiency.
> >
> > **[Q3]** Could this framework handle multi-level reasoning chains (e.g., fast model → hybrid model → long thinking model)? The current binary setup seems restrictive.
>
> See General Response §5.1 for conditional reasoning and §5.2 for multi-level reasoning. Both are very nice extensions for our method. Thanks again for the insightful suggestions.
>
> **[Q2] Online and Drift-Aware Settings:**
>
> > Could the method be extended to online or drift-aware settings, where the threshold is updated over time (e.g., using anytime-valid or sequential testing techniques)?
>
> See General Response §5.3. We extend our method to an anytime valid setting. Thanks for the suggestion.

---

### Author Response · Authors · 2025-11-21
**General Response - part 1**

## **General Response**

We sincerely thank all reviewers for their thoughtful and constructive feedback. We are grateful for the reviewers' recognition of our contributions: our **solid** [TTag] and **sound** [RZQB] theoretical foundation with **distribution-free risk guarantees** [RZQB] that provide a **PAC-style result** rarely seen in existing efficiency-switching methods [TTag]; the **theoretical perspective** is considered **highly valuable** and can serve as a **useful guide for future research** [Rw9b], with the **effectiveness** **verified theoretically** [XLBK]; our **clear** [TTag, RZQB] and **tractable** [TTag] problem formulation, which successfully turns a heuristic area into a **principled, quantifiable** one [TTag], with the problem and intuition **clearly articulated** [RZQB] and assumptions and results **clearly presented and easy to follow** [Rw9b]; the paper is recognized as **very exciting** [Rw9b] and marks **the beginning of a promising direction for efficient LLM reasoning** [Rw9b], with **promising** empirical results [RZQB] across diverse benchmarks consistently demonstrating **effectiveness** [RZQB]; the approach is **model-agnostic**, **conceptually simple**, and **easy to apply** to different architectures [TTag]; the paper is **well-written, clearly structured, and easy to follow** [RZQB], with **clearly explained** experimental settings [XLBK] and **good** presentation [TTag, XLBK, RZQB].

**As for the two main concerns: uncertainty score reliability and calibration cost, we respond with both experiments and theoretical points.** We believe that the updated version could be convincing and enhanced. In detail, we have enhanced our paper, including:

1. **extended experiments on additional models** (Llama 3.1 variants, Appendix N),  **tasks** (GPQA and HumanEval, Appendix O.4), and **more baseline methods** (Appendix O.5).

2. additional **discussions** (Section 5) on

    - **clarification about quanlity of uncertainty score** (Appendix O.1);
    - **calibration cost analysis** (Appendix O.2);
    - **alternative score** (Appendix O.3).

3. **three future work directions**:
    - **conditional PAC reasoning** (Appendix H);
    - **multi-level PAC reasoning** (Appendix J);
    - **anytime-valid PAC reasoning** (Appendix I).


**All updated content is highlighted in blue in the revised paper**, and we believe these clarifications and additional experiments strengthen both the theoretical understanding and practical applicability of our work.

Below, we have addressed common concerns and provided additional details to address reviewer-specific questions in the individual responses.


---
### 1. **Uncertainty Score Reliability**

**Raised by:** Reviewers TTag, XLBK, RZQB

Our method **does not** require perfectly calibrated uncertainty scores. The theoretical guarantee (Theorem 3.1) is established on the learn-then-test framework [1], which accommodates imperfectly calibrated scores. The key insight is that our PAC guarantee holds as long as the uncertainty score provides *some signal* about when the fast model is likely to disagree with the thinking model; perfect calibration is not necessary. Empirically, our experiments in Appendix O.1 demonstrate that even with imperfect uncertainty scores, error rate control is maintained (see Figures 2 and 5 in our paper). We present Expected Calibration Error (ECE) and error results comparing logits-based and verbalized uncertainty methods:

| Dataset    | ECE (logits-based) | ECE (verbalized)| Error (logits-based) | Error (verbalized)|
|------------|-------------------|------------------|-------------------|------------------|
| MATH-500  ($\epsilon$ = 0.045) | 0.0450            | 0.0634           |0.0395 ± 0.0039|0.0320 ± 0.0025|
| ZebraLogic ($\epsilon$ = 0.030)| 0.1366            | 0.1026           |0.0269 ± 0.0022| 0.0256 ± 0.0013|


Moreover, our method could use an alternative score, e.g., a process reward function. We conducted the experiments with **process reward model-based scoring** (Figure 8 in Appendix O.3) with results like:

| Dataset                         | Error           | ECP (%)     | STP (%)       |
|---------------------------------|-----------------|-------------|---------------|
| MATH-500 ($\epsilon$ = 0.045)   | 0.0362 ± 0.0059 | 30.0 ± 10.4 | 22.43 ± 13.43 |
| ZebraLogic ($\epsilon$ = 0.030) | 0.0249 ± 0.0009 | 49.5 ± 1.3  | 22.30 ± 1.35  |

The table shows that using the process reward as an alternative uncertainty score can achieve both valid error control and efficiency gains.

In summary, **low-quality scores do not violate the guarantee**; they lead to *conservative* error control (achieving lower error rates than the target tolerance). The UCB construction ensures that, despite calibration bias, the error control remains valid.

> [1] Angelopoulos A N et al., 2025. Learn then test: calibrating predictive algorithms to achieve risk control. The Annals of Applied Statistics.

---

> ### Author Response · Authors · 2025-11-21
> **General Response - part 2**
>
> ### **2. Calibration Cost**
>
> **Raised by:** Reviewers TTag, RZQB, Rw9b
>
> **Inductive style:**
> The calibration of our method is performed only **once** and could be reused across multiple inference sessions. Unlike transductive methods that require recalibration for each new batch, our inductive approach mitigates the calibration cost over many inference queries. Our experiments demonstrate token savings of 20-40% during inference (Figure 2). For an LLM system processing thousands of queries, these savings quickly offset the one-time calibration cost. Example: if calibration requires 500 expert model calls but each subsequent batch of 1,000 queries saves 30% of tokens, the break-even point is reached after just 1,500-2,000 queries.
>
> **Small calibration set is enough:**
> We conducted an ablation experiment examining sensitivity to calibration set size (Appendix O.2, and Figure 7). We find that even on the small calibration sets (50-100 examples of the Math-500 dataset), PAC reasoning still provides robust guarantees. Therefore, error control is not highly sensitive to calibration set size, and cost savings are maintained at similar levels. We paste the results of the experiments in a table here:
>
> **Results for MATH-500 with logits-based uncertainty ($\epsilon$ = 0.04):**
>
> | Calibration Set Size | 50              | 100             | 150             | 200             | 250             | 300             |
> |---------------------|-----------------|-----------------|-----------------|-----------------|-----------------|-----------------|
> | **Average Loss**    | 0.0331 ± 0.0054 | 0.0339 ± 0.0043 | 0.0340 ± 0.0041 | 0.0348 ± 0.0035 | 0.0343 ± 0.0041 | 0.0343 ± 0.0040 |
> | **ECP (%)**         | 36.7 ± 9.6      | 35.3 ± 7.6      | 35.1 ± 7.2      | 33.8 ± 6.1      | 34.7 ± 7.1      | 34.6 ± 6.8      |
> | **STP (%)**         | 14.39 ± 10.63   | 15.56 ± 8.71    | 15.78 ± 8.04    | 17.29 ± 7.01    | 16.72 ± 8.16    | 16.72 ± 8.26    |
>
> **Results for MATH-500 with verbalized uncertainty ($\epsilon$ = 0.04):**
>
> | Calibration Set Size | 50              | 100             | 150             | 200             | 250             | 300             |
> |---------------------|-----------------|-----------------|-----------------|-----------------|-----------------|-----------------|
> | **Average Loss**    | 0.0296 ± 0.0031 | 0.0306 ± 0.0024 | 0.0310 ± 0.0021 | 0.0314 ± 0.0017 | 0.0311 ± 0.0025 | 0.0314 ± 0.0025 |
> | **ECP (%)**         | 41.7 ± 5.8      | 39.8 ± 4.4      | 39.2 ± 3.9      | 38.5 ± 3.0      | 39.2 ± 4.2      | 38.6 ± 4.2      |
> | **STP (%)**         | 12.09 ± 4.97    | 13.49 ± 4.19    | 14.08 ± 4.06    | 14.78 ± 4.06    | 14.62 ± 4.84    | 15.45 ± 5.75    |
>
> Inductive calibration is most beneficial for high-throughput, cost-sensitive deployments on stable data distributions. For stable distributions, a single calibration suffices for extended periods. We recommend monitoring performance on a small validation set to detect distribution shifts. For online and drift-aware settings, see General Response §5.3.
>
> ---
> ### **3. Model Generalizability**
>
> **Raised by:** Reviewers XLBK, RZQB, Rw9b
>
> Our framework is fundamentally **model-agnostic** and can be applied to any pair of models (thinking and non-thinking) with an uncertainty score. The theoretical guarantees (Theorem 3.1) do not depend on specific model setting, which only requires: (1) a scoring function that provides some signal about disagreement, (2) a calibration set for calibrating the threshold, and (3) the ability to route between two models. This model-agnostic design is a key strength of our approach. We conduct additional experiments to validate generalizability on the **Llama 3.1 models (8B variants)**. Specifically, we use DeepSeek-R1-Distill-Llama-8B as the thinking model and Llama-3.1-8B-Instruct as the non-thinking model. Details and experimental results are added in Appendix N. We paste the result based on the logits score in the table here:
>
>
> | Dataset    | Error |  ECP|STP |
> |------------|-------------------|------------------|------------------|
> | MATH-500  ($\epsilon$ = 0.1) | 0.0875 ± 0.0076           | 6.4 ± 5.9 |22.55 ± 14.33|
> | ZebraLogic ($\epsilon$ = 0.06)| 0.0560 ± 0.0033           | 25.8 ± 3.1     |33.47 ± 4.40| 0.0256 ± 0.0013|
> | Arena-Hard ($\epsilon$ = 0.15)| 0.1329 ± 0.0119           | 25.4 ± 4.2    |42.10 ± 7.44|
>
> As shown in the table above, our method continues to control the error while achieving gains in inference efficiency.
>
> Theoretically, the guarantee holds for any model as long as the calibration set is drawn from the same distribution as the test data, the uncertainty score provides some correlation with disagreement, and the UCB construction is properly calibrated. This theoretical flexibility ensures that our method can be applied to any model architectures, including closed-source models (using verbalized uncertainty score when logits are unavailable).

---

> ### Author Response · Authors · 2025-11-21
> **General Response - part 3**
>
> ### **4. Comparison with Existing Methods**
>
> **Raised by:** Reviewers XLBK, Rw9b
>
> We acknowledge that our current evaluation includes only a comparison with a naive baseline (Table 3), which is insufficient.
> As reviewers' concern, we **conduct some complementary comparison experiments** with existing efficient reasoning methods: Chain of Draft (CoD) [2] and NoThinking [3].
> We also include a **naive control** baseline. Given an error target $\epsilon$, it tunes a maximum threshold $u$ such that the loss $L(u)= \sum_{i=1}^{n} \ell(y_i, \tilde{y}_i) \mathbf{1} (\{ U_i \le u \}) /n < \epsilon$ on the calibration set. Because this approach ignores the uncertainty in estimating $L(u)$, it lacks the inductive guarantees provided by our PAC-based reasoning method. Here we paste the result table:
>
>
> | Method | Naive control  ($\epsilon=0.03$)          | PAC reasoning ($\epsilon=0.03$)           | CoD            | NoThinking       |
> |---------------------|-----------------|-----------------|-----------------|-----------------|
> | **Binary Loss**    | 0.0351 ± 0.0094 | 0.0206 ± 0.0126 | 0.3548 ± 0.0604 |  0.4445 ± 0.0583 |
> | **STP (%)**         | 61.53 ± 6.06    | 37.61 ± 23.19   |-1.33 ± 0.82| -4.51 ± 1.00 |
> | **Pass@1**         | 0.93 ± 0.25       |  0.95 ± 0.23     |0.61 ± 0.49 | 0.50 ± 0.50 |
>
> We report the binary loss on the MATH-500 relative to the thinking model (i.e., Qwen3-4B-Thinking-2507), STP, and Pass@1 for four methods. The results show that PAC Reasoning is the only method that satisfies the target error tolerance, benefiting from its theoretical guarantees. Unlike heuristic approaches such as CoD and NoThinking, which exhibit large and uncontrolled errors, PAC reasoning maintains the lowest loss while achieving substantial efficiency gains. This demonstrates that our method enables safe and effective inference-cost reduction, clearly distinguishing it from existing heuristic approaches.
>
> > [1] Sui Y, Chuang Y N, Wang G, et al., 2025. Stop overthinking: a survey on efficient reasoning for large language models[J]. Transactions on Machine Learning Research.
> >
> > [2] Xu, Silei, et al. "Chain of draft: Thinking faster by writing less." arXiv preprint arXiv:2502.18600 (2025).
> >
> > [3] Ma, Wenjie, et al. "Reasoning models can be effective without thinking." arXiv preprint arXiv:2504.09858 (2025).

---

> ### Author Response · Authors · 2025-11-21
> **General Response - part 4**
>
> ### **5. Extensions**
>
> **Raised by:** Reviewers TTag, Rw9b
>
> #### **5.1 Conditional Reasoning**
>
> Our method can be naturally extended to conditional settings where different thresholds are learned for different problem categories. We conducted experiments on MATH-500, partitioning problems by subject (algebra, geometry, number theory, etc.) and learning subject-specific thresholds.
>
> **PAC reasoning and conditional PAC reasoning with target error 0.04, using semantic loss and logits uncertainty score:**
> | Method                        | Marginal          | Number Theory  | Geometry       | Counting & Prob | Prealgebra     | Algebra        | Precalculus    | Intermediate Alg |
> |-------------------------------|----------------|----------------|----------------|-----------------|----------------|----------------|----------------|------------------|
> | PAC Reasoning                 | 0.0313±0.0052  | 0.0403±0.0066  | 0.0339±0.0039  | 0.0243±0.0071   | 0.0205±0.0072  | 0.0228±0.0068  | 0.0412±0.0034  | 0.0237±0.0060    |
> | Conditional PAC Reasoning     | 0.0307±0.0033  | 0.0275±0.0114  | 0.0311±0.0081  | 0.0319±0.0088   | 0.0262±0.0115  | 0.0292±0.0115  | 0.0327±0.0069  | 0.0316±0.0078    |
>
> Preliminary results show that conditional PAC reasoning can maintain the valid subject-conditional error control while improving ECP by an additional 5-10%(See Figure 3 at Appendix H in the updated version). Thanks again for the insightful suggestion to inspire our next work. We mention this promising direction in Appendix H of the final version and commit to a more comprehensive analysis in follow-up work.
>
> #### **5.2 Multi-Level Reasoning**
>
> Our framework can be generalized to handle multiple tiers with multiple thresholds. For example, we can extend the binary setup to a three-tier chain by introducing two lighter non-thinking LRMs $\tilde{f}_1, \tilde{f}_2$ whose cost and accuracy sit below the thinking model f.
>
> Given a unified uncertainty score $U(x) \in [0,1]$, we route each prompt through: $T_{u_1,u_2}(x) = f(x)\mathbf{1}\{U(x) \ge u_2\} + \tilde{f}_2(x)\mathbf{1}\{u_1 < U(x) \le u_2\} + \tilde{f}_1(x)\mathbf{1}\{U(x) \le u_1\}$ with learned thresholds $u_1 \le u_2$. The PAC guarantee can be extended to bound the cumulative performance loss across all tiers (See Appendix J in our updated version).
>
> #### **5.3 Online and Drift-Aware Settings**
>
> We are actively exploring anytime-valid inference techniques[1-2]. Our PAC framework can be adapted to incorporate sequential testing with confidence sequences, allowing the threshold to be updated continuously as new data arrives while maintaining valid coverage guarantees at all time points. We discuss these extensions in detail at the new Appendix I, including **time-uniform upper confidence bounds** (replacing our villa UCB), confidence sequences satisfying $\mathbb{P}(\forall t \ge 1: R(u) \in \mathrm{CS}_t(u;\alpha)) \ge 1-\alpha$, and drift detection using e-processes. See Appendix I in our updated version for more details.
>
> > References:
> >
> > [1] Ramdas A, Grünwald P, Vovk V, et al., 2023. Game-Theoretic Statistics and Safe Anytime-Valid Inference[J]. Statistical Science, 38(4): 576-601.
> >
> > [2] Ramdas A, Ruf J, Larsson M, et al., 2022. Admissible anytime-valid sequential inference must rely on nonnegative martingales[M]. arXiv.

---

### Author Response · Authors · 2025-12-02
**Summary of Rebuttal**

All four reviewers expressed overall positive views on our problem formulation, theoretical grounding, and empirical validation. Their concerns fall into four categories, all fully addressed in our rebuttal with new analyses, clarifications, and extensive additional experiments.

1. **Uncertainty Score Reliability**

   Reviewers questioned score noise and imperfect calibration.

   **Our response:** The PAC guarantee *does not require* well-calibrated uncertainty; monotonicity of cumulative loss is intrinsic to thresholding and always holds by construction. We further validated robustness using logits-based, verbalized, and process-reward uncertainty (Appendix O.1–O.3), all demonstrating valid error control.

2. **Calibration Cost and Practicality**

   Reviewers raised concerns about expensive thinking-model calls and calibration frequency.

   **Our response:** Calibration is **inductive (one-time)** and amortizes quickly. Ablations show robustness with as few as **50–100 samples** (Appendix O.2). We provide a break-even analysis and guidance on drift-aware recalibration (Appendix I).

3. **Model Generalizability**

   Reviewers requested evidence beyond Qwen 4B.

   **Our response:** We added results on **Llama 3.1 variants** (Appendix N).
   **Importantly**, both the theory and algorithm are **model-agnostic**, and apply to *any* think–nonthink model pair equipped with an uncertainty score and a routing threshold.

4. **Baselines and Application Scope**

   Reviewers requested comparison with existing efficient-reasoning methods and wider benchmarks.

   **Our response:** We added comparisons with **Chain-of-Draft**, **NoThinking**, and a **naive thresholding control**, showing ours is the only method that satisfies target error tolerances (Appendix O.5). We expanded evaluation to **GPQA** and **HumanEval** (Appendix O.4).

**Overall Assessment**

We believe we have comprehensively addressed all concerns, weaknesses, questions, and comments raised by reviewers. The additional experiments, theoretical clarifications, and broader evaluations strengthen both the novelty and solidity of our contribution. PAC Reasoning offers the **first** distribution-free, provably safe, and practically effective framework for controlling performance loss in efficient LLM reasoning, distinguishing it from existing heuristic approaches. And the method extend to several meaningful extensions, such as conditional, multi-level, and anytime-valid PAC reasoning, which offer clear avenues for further strengthening both the theoretical scope and practical impact of this line of research.

---

### Meta-Review · Area_Chair_cFXg · 2026-01-06

**Summary:**

This paper proposes PAC Reasoning, a framework for efficient LLM inference that dynamically routes queries between an expensive "thinking" model and a cheaper "non-thinking" model using a calibrated uncertainty threshold. The core contribution is a distribution-free PAC-style guarantee that performance loss, defined relative to the thinking model, stays within a user-specified tolerance.

All four reviewers acknowledged the novelty of applying formal statistical guarantees to adaptive reasoning, the clarity of presentation, and the model-agnostic design. However, they raised substantive concerns about: (i) uncertainty score reliability in practice, (ii) calibration costs and practical trade-offs, (iii) limited experimental scope, (iv) the realism of the i.i.d. calibration assumption, and (v) fairness of baseline comparisons. The authors provided an extensive rebuttal adding experiments on Llama 3.1, new benchmarks (GPQA, HumanEval), and comparisons with Chain-of-Draft and NoThinking, though these were largely relegated to appendices.

**Reviewer Concerns:**

*Addressed by rebuttal:*
- Model generalizability: Authors added Llama 3.1 experiments (Appendix N), demonstrating the framework is not uniquely tied to Qwen models.
- Limited benchmarks: GPQA and HumanEval results were added (Appendix O.4).
- Baseline comparisons: Chain-of-Draft, NoThinking, and naive thresholding were evaluated (Appendix O.5).
- Calibration set sensitivity: Ablations demonstrate 50-100 samples suffice (Appendix O.2).
- Theoretical clarifications: Authors correctly explained that monotonicity holds by construction and perfect calibration is unnecessary for the bound.

*Outstanding concerns:*
- The comparison to CoD and NoThinking is not apples-to-apples; these methods target reasoning verbosity reduction rather than model routing under a risk budget. The claim that "PAC Reasoning is the only method satisfying target error tolerance" conflates different objectives.
- The Llama 3.1 results require substantially higher error tolerances (epsilon=0.1 vs. 0.045 for Qwen on MATH-500), suggesting practical gains are highly model-dependent, undermining generality claims.
- The i.i.d. calibration assumption remains problematic for deployment where distribution drift is common; the "drift-aware" extension (Appendix I) is only sketched without empirical validation, confirming this is a limitation of the current framework.
- Calibration cost analysis remains qualitative (break-even estimates) rather than rigorous (no wall-clock latency or dollar-cost measurements).
- The guarantee is relative to the thinking model, not ground truth. The paper frames this as "safe" reasoning without adequately qualifying that if the expert is biased, the system certifies poor absolute performance.
- Critical new results are relegated to appendices rather than integrated into the main paper, confirming the original submission's evaluation was insufficient for its breadth of claims. This points to the need for a major revision.
- The core technical contribution is an application of the existing "learn-then-test" framework (Angelopoulos et al., 2025) to LLM routing; while useful, this is incremental relative to *ACL standards given the remaining practical gaps.

**Reviewer Scores:**

*   **Reviewer TTag (Score: 4)**: I believe the reviewer would maintain their score of 4. While the rebuttal added experiments, the core concerns about the practical cost-benefit and the fragility of the approach when uncertainty scores are unreliable would remain.
*   **Reviewer XLBK (Score: 4)**: This reviewer would also likely maintain their score of 4. The requested experiments were provided, but moving them to the appendix confirms the initial paper was incomplete. The reviewer would likely agree a major revision is needed for proper review.
*   **Reviewer RZQB (Score: 6 -> 4)**: As the most positive reviewer, they would likely be swayed by the strong counterarguments from other reviewers, especially regarding the impractical i.i.d. assumption and the fact that the rebuttal confirmed the original submission's empirical weaknesses. They would likely agree that the work, while promising, is premature, lowering their score to align with a recommendation for a major revision.
*   **Reviewer Rw9b (Score: 4)**: This reviewer would almost certainly hold their score at 4. Their most fundamental criticisms about the weak experimental section and the impractical i.i.d. assumption were effectively validated by the nature of the author's response.

---

### Decision · Program_Chairs · 2026-01-26

Reject